# Reversible surface modifications of functional proteins for accelerated cytosolic delivery via cell-penetrating peptide clusters

Xiao Hua[1,2,3,5], Yanyan Guo[3,5], Pincheng Li[1,5], Yu Wang[1,5], Xiaona Han[1], Junyou Chen[1], Junjiang Li[1], Guo-Chao Chu[1,2], Jing Shi[3], Lei Liu [2] ✉ & Yi-Ming Li [1,4] ✉

A long-standing goal in biomedical research is to label and manipulate intracellular targets, which could be achieved through the cytosolic delivery of exogenous functional proteins. The development of Tat clusters has advanced the nontoxic intracellular delivery of functional antibodies at low concentrations, but the variety of proteins that can be successfully delivered remains limited. Here, we find that by simply reversibly modifying the surface of functional proteins with anionic peptide patches, various protein cargoes (which are normally difficult to deliver) can be delivered into living cells by synergetic electrostatic interactions with the cationic cell-penetrating peptide clusters TAT$_3$. To demonstrate the applicability of this approach, we successfully deliver functional proteins with widely varying molecular weights (~1.5 kDa to 430 kDa) and isoelectric points (less than 5 to greater than 9) into the cytosol of cells. By exploiting this method, we also achieve protein delivery in plant tissues, which is more challenging due to the presence of intact plant cell walls. This strategy is further applied for the cytosolic delivery of synthetic protein probes carrying posttranslational modifications (PTMs), which can aid in in situ mapping of the intracellular PTM-mediated interactome. Overall, this strategy is expected to enrich cytosolic protein delivery technology and help to repurpose a wide range of customized and therapeutic proteins for emerging intracellular applications.

The ability to deliver exogenous proteins into living cells is important for a wide range of applications in biomedical research, e.g., profiling intracellular protein–protein interactions (PPIs) using synthetic protein probes equipped with photoreactive warheads, visualizing and targeting organelles using fluorescent protein-labeled cargoes, and detecting and localizing antigens in cells and tissues using exogenous monoclonal antibodies[1–3]. To overcome the poor cell membrane permeability of proteins, several ingenious strategies have been explored for delivering proteins into the cytosol of cells[4–6]; among these systems, cell-penetrating peptide (CPP)-mediated delivery has emerged as a prominent carrier system[7,8]. CPPs can deliver a wide range of protein cargoes, such as functional enzymes, small antibody fragments (nanobodies) and fluorescent proteins, into living cells[9–11]. However, cargo proteins must be added at relatively high

[1]School of Food and Biological Engineering, Engineering Research Center of Bioprocess, Ministry of Education, Key Laboratory of Animal Source of Anhui Province, Hefei University of Technology, Hefei 230009, China. [2]Tsinghua-Peking Center for Life Sciences, Ministry of Education Key Laboratory of Bioorganic Phosphorus Chemistry and Chemical Biology, Center for Synthetic and Systems Biology, Department of Chemistry, Tsinghua University, Beijing 100084, China. [3]Department of Chemistry, University of Science and Technology of China, Hefei 230026, China. [4]Beijing Life Science Academy, Beijing 102200, China. [5]These authors contributed equally: Xiao Hua, Yanyan Guo, Pincheng Li, Yu Wang. ✉e-mail: lliu@mail.tsinghua.edu.cn; ymli@hfut.edu.cn

concentrations (>10 μM) to guarantee delivery efficiency, which is either inconvenient or impossible due to solubility or toxicity[10,12,13].

A recent landmark finding was that the strategic assembly of the highly positively charged CPP Tat into multimeric clusters greatly improved its cellular uptake efficiency. Building upon this finding, Tietz et al. developed a family of trimeric Tat that enables the nontoxic intracellular delivery of functional antibodies and antibody fragments at low concentrations (e.g., 1-2 μM) by simple incubation of Tat clusters with protein cargoes[14]. Encouraged by these important discoveries, we became interested in the cytosolic delivery of other types of functional proteins, such as fluorescent proteins for targeted labeling, E2-ubiquitin (Ub) conjugates for in situ mapping of PPIs of Ub ligases, and bioactive enzymes that can degrade intracellular RNA. However, we and other researchers have found that after coincubation with trimeric Tat, only a small variety of proteins can be delivered into cells at low concentrations[14]; that is, a wide range of protein cargoes with different isoelectric points and molecular weights are difficult to deliver via the Tat cluster strategy. One possible reason is that these protein cargoes lack favorable interactions with trimeric Tat.

Here, we found that by simply modifying the surface of functional proteins with anionic peptide patches, protein cargoes that are otherwise difficult to deliver could be readily delivered into living cells at ~1.5 μM by synergetic electrostatic interactions with the cationic CPP TAT3. Motivated by these observations, we developed a reversible anionic patch modification-based approach to enable the intracellular delivery of a broad variety of functional proteins, including antibodies, synthetic protein probes bearing posttranslational modifications (PTMs), fluorescent proteins, and functional enzymes, whose molecular weights range from ~1.5 kDa to 430 kDa and isoelectric points range from less than 5 to greater than 9 (Fig. 1). Upon cellular uptake, the anionic patch spontaneously dissociates from the protein, thereby releasing native cargoes. This strategy was also applied for the cytosolic delivery of protein cargoes into intact plants, which is more challenging to deliver due to the presence of plant cell walls. Finally,

the utility of this strategy was exemplified by the cytosolic delivery of synthetic protein probes, such as the E2-Ub photoaffinity probe, which helps in in situ mapping of the PTM-mediated intracellular interactome.

## Results

### Anionic surface modification achieves cellular uptake of UbcH7_86C

Owing to our enduring interest in the use of E2 probes to identify their partner E3s (Ub E3 ligases)[15], we began this work with the cytosolic delivery of a commonly used E2 enzyme, UbcH7 (pI: 8.87, MW: 18.7 kDa)[16]. The CPP cR10 was selected for conjugation[17] to UbcH7_86C-T (a UbcH7 mutant retaining only the catalytic cysteine at position 86 (Supplementary Fig. 3a), which was coupled to fluorophore NHS-TAMRA (T)) via a disulfide bond. However, we observed significant precipitation during the cR10 conjugation (Supplementary Fig. 3b) disfavoring its further implementation. We then turn to the co-incubation strategy with TAT peptides and test three types of TAT: trimeric cyclic TAT (tri-cTat, developed by Tietz et al.[14]), trimeric linear TAT (TAT3, derived from Brock et al.[18]) and mono-TAT (commonly used CPP) (Supplementary Fig. 1a-1c and 3c). These TATs were each mixed with 1.5 μM UbcH7_86C-T and then incubated with HepG2 cells for 30 minutes. Confocal laser scanning microscopy (CLSM) analysis revealed no detectable diffuse red fluorescence in cells treated with 1 μM or 3 μM TAT trimers (tri-cTat and TAT3) or equivalent molar concentrations of mono-TAT (3 μM and 9 μM) (Supplementary Fig. 3d-3f). Therefore, both previous approaches, covalent conjugation with cR10 and coincubation with TAT, failed to deliver UbcH7_86C into cells.

One plausible explanation for the failed delivery of UbcH7_86C by co-incubation with TAT may due to its positive charge at physiological pH (pI: 8.87), which counteracts with the positively charged TAT peptides. To reduce the positive charge of UbcH7, we designed and synthesized an acidic amino acid-rich peptide E3D2 (sequence: PEDEDELAC) (Supplementary Fig. 1d). The cysteine residue in E3D2

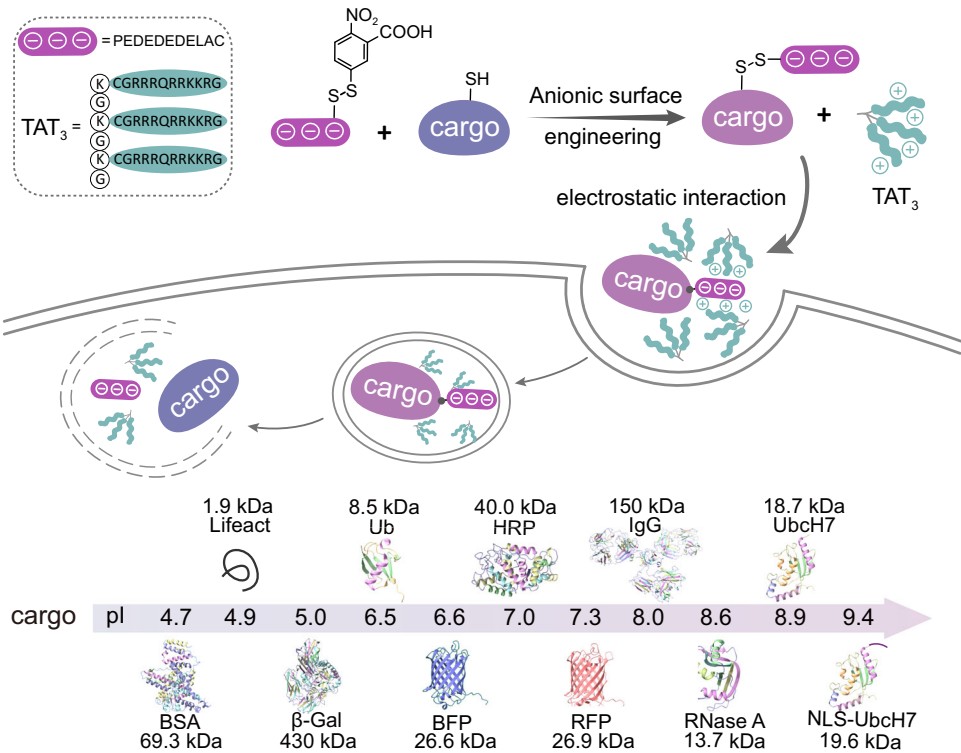

**Fig. 1 | Schematic diagram of the strategy in this article.** Synergizing anionic surface modification and electrostatic interactions with TAT3 for protein cytosolic delivery. Protein cargoes whose molecular weights range from ~1.5 kDa to 430 kDa and isoelectric points range from less than 5 to greater than 9 can be delivered with this strategy.

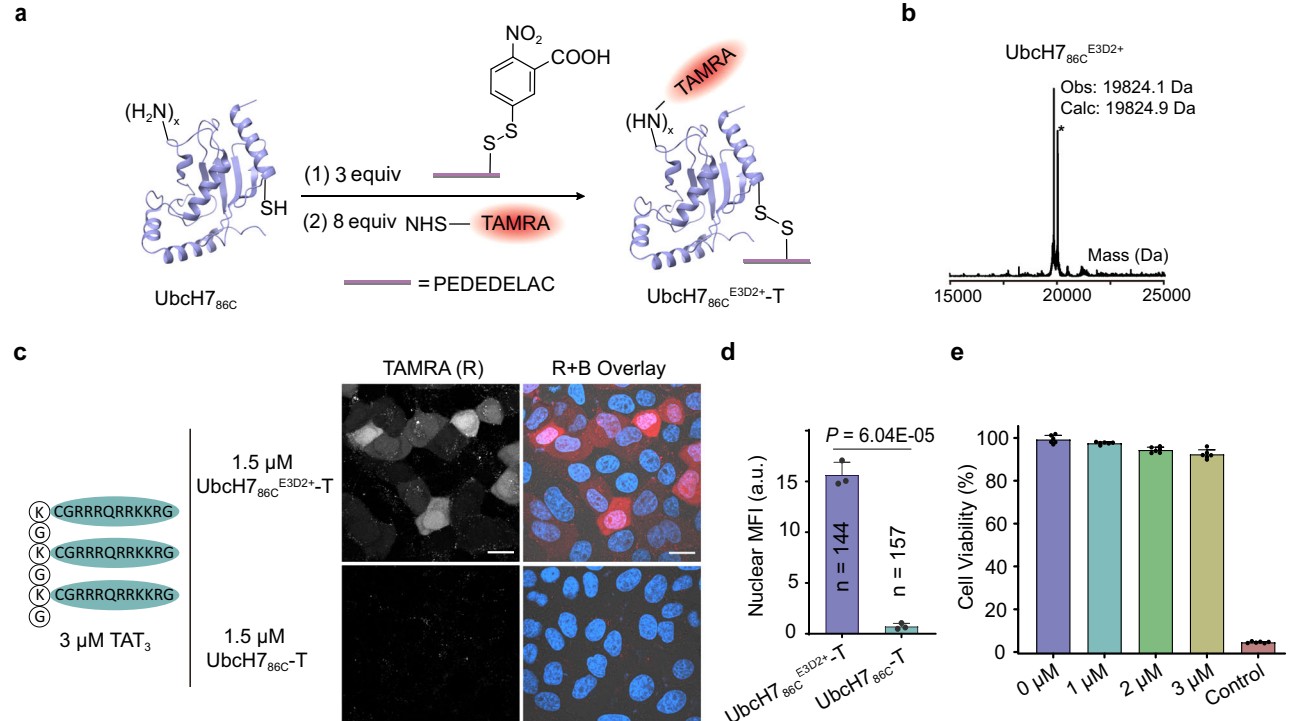

**Fig. 2 | Cellular uptake of E3D2 peptide-modified UbcH7$_{86C}$. a** Schematic depiction of E3D2 peptide-modified UbcH7$_{86C}$. **b** Deconvoluted mass characterization of UbcH7$_{86C}$$^{E3D2+}$. "*" corresponds to a molecular weight of 20002.1 Da, denoting 178 Da greater than the desired mass is attributed to the modified product of gluconoylated[41] UbcH7$_{86C}$ during protein expression. **c** Confocal microscopy images of HepG2 cells treated with 1.5 μM UbcH7$_{86C}$$^{E3D2+}$-T or UbcH7$_{86C}$-T in the presence of TAT$_3$ for 30 minutes at 37 °C. "B" and "R" stand for Hoechst (blue) and TAMRA (red) signals, respectively. The experiment was repeated three times. Scale bars, 20 μm. **d** Quantification of the nuclear TAMRA mean fluorescence intensity

(MFI) of the cells treated in **c**. MFI is expressed in arbitrary units (a.u.). The results are the average of three biological replicates ($n = 144$ for UbcH7$_{86C}$$^{E3D2+}$-T group and $n = 157$ for UbcH7$_{86C}$-T group) and presented as the mean ± standard deviation. $P$-value was calculated via two-sided $t$ test. **e** Cell viability detected by a CCK-8 assay after treating HepG2 cells with 0, 1, 2, or 3 μM TAT$_3$ for 30 minutes in serum-free DMEM. Treatment with 0.64% phenol was used as a control. The results are the average of three biological replicates The data are presented as the mean ± standard deviation. Source data are provided as a Source Data file.

was first activated by 5,5-dithiobis (2-nitrobenzoic acid) (DTNB) reagent (Supplementary Fig. 1e), followed by a disulfide exchange reaction with UbcH7$_{86C}$ in phosphate-buffered saline (PBS, pH 7.4) (Fig. 2a). Mass spectrometry analysis verified the almost complete conversion of UbcH7$_{86C}$ to the mono-E3D2 modified adduct (UbcH7$_{86C}$$^{E3D2+}$) within 1 hour (Fig. 2b), which was subsequently purified by fast protein liquid chromatography (FPLC). The UbcH7$_{86C}$$^{E3D2+}$ conjugate was fluorescently labeled with NHS-TAMRA, and the labeled product (UbcH7$_{86C}$$^{E3D2+}$-T) was mixed with three different TATs and then co-incubated with HepG2 cells for 30 minutes. Diffuse TAMRA red fluorescence was observed in cells treated with 1.5 μM UbcH7$_{86C}$$^{E3D2+}$-T and 3 μM TAT$_3$ (Fig. 2c). Z-stack imaging further confirmed the cytoplasmic and nuclear distribution of the fluorescence (Supplementary Fig. 4a). Nuclear mean fluorescence intensity (MFI) was used here because it can distinguish proteins that have successfully escaped into the cytosol and those that are trapped within endosomes[11]. MFI analysis revealed that the MFI of HepG2 cells co-incubated with UbcH7$_{86C}$$^{E3D2+}$-T and TAT$_3$ was as high as 16 compared to the unmodified UbcH7$_{86C}$-T (MFI = 1) (Fig. 2d). Cytotoxicity assessment (CCK-8 assay)[19] showed >95% cell viability at TAT$_3$ concentrations between 1-3 μM (Fig. 2e). These results demonstrate that E3D2 surface modification enables efficient intracellular delivery of UbcH7$_{86C}$ when mixed with TAT$_3$ with minimal cytotoxicity in HepG2 cells. In comparison, when the same concentration of UbcH7$_{86C}$$^{E3D2+}$-T was delivered, although some punctate red fluorescence[20] was observed in cells co-incubated with 3 μM tri-cTat, the cells had simultaneously exhibited significant membrane rupture, whereas little red

fluorescence was observed in cells co-incubated with 9 μM mono-TAT (Supplementary Fig. 4b).

## Reversible surface modification enables efficient cellular uptake of UbcH7$_{86C}$ with minimal cytotoxicity

To investigate the effect of the number of acidic amino acids in peptide sequences on UbcH7$_{86C}$ delivery efficiency, we designed and synthesized five peptides with different numbers of acidic amino acids (Supplementary Fig. 1d, 1f-1i), which were subsequently conjugated to UbcH7$_{86C}$ using the above protocol (Fig. 2a). Fluorescence imaging and quantitative analysis demonstrated the successful intracellular delivery of all modified cargoes by co-incubation with 3 μM TAT$_3$ (Fig. 3a and Supplementary Fig. 5), in which UbcH7$_{86C}$$^{E4D3+}$-T exhibited the highest nuclear MFI. Despite containing more acidic amino acids in their sequences, UbcH7$_{86C}$$^{E6D5+}$-T and UbcH7$_{86C}$$^{E5D4+}$-T showed approximately 50% reduction in nuclear MFI compared to UbcH7$_{86C}$$^{E4D3+}$-T (Fig. 3b). Based on these results, we selected the E4D3 sequence (PEDEDEDELAC) for the surface modification of UbcH7$_{86C}$ (the adduct was designated UbcH7$_{86C}$$^+$).

Further concentration gradient experiments revealed that UbcH7$_{86C}$$^+$-T in the concentration range from 0.5 to 9 μM could be delivered into cells by co-incubation with 3 μM TAT$_3$ (Supplementary Fig. 6a), with nuclear MFI showing a linear correlation with incubation concentrations (Fig. 3c). Cell viability remained >95% at all concentrations tested (Supplementary Fig. 6b). We then prepared E4D3-modified UbcH7 by disulfide bonds at either the N-terminus (UbcH7$_{17C}$) or C-terminus (UbcH7$_{137C}$). Both cargoes showed

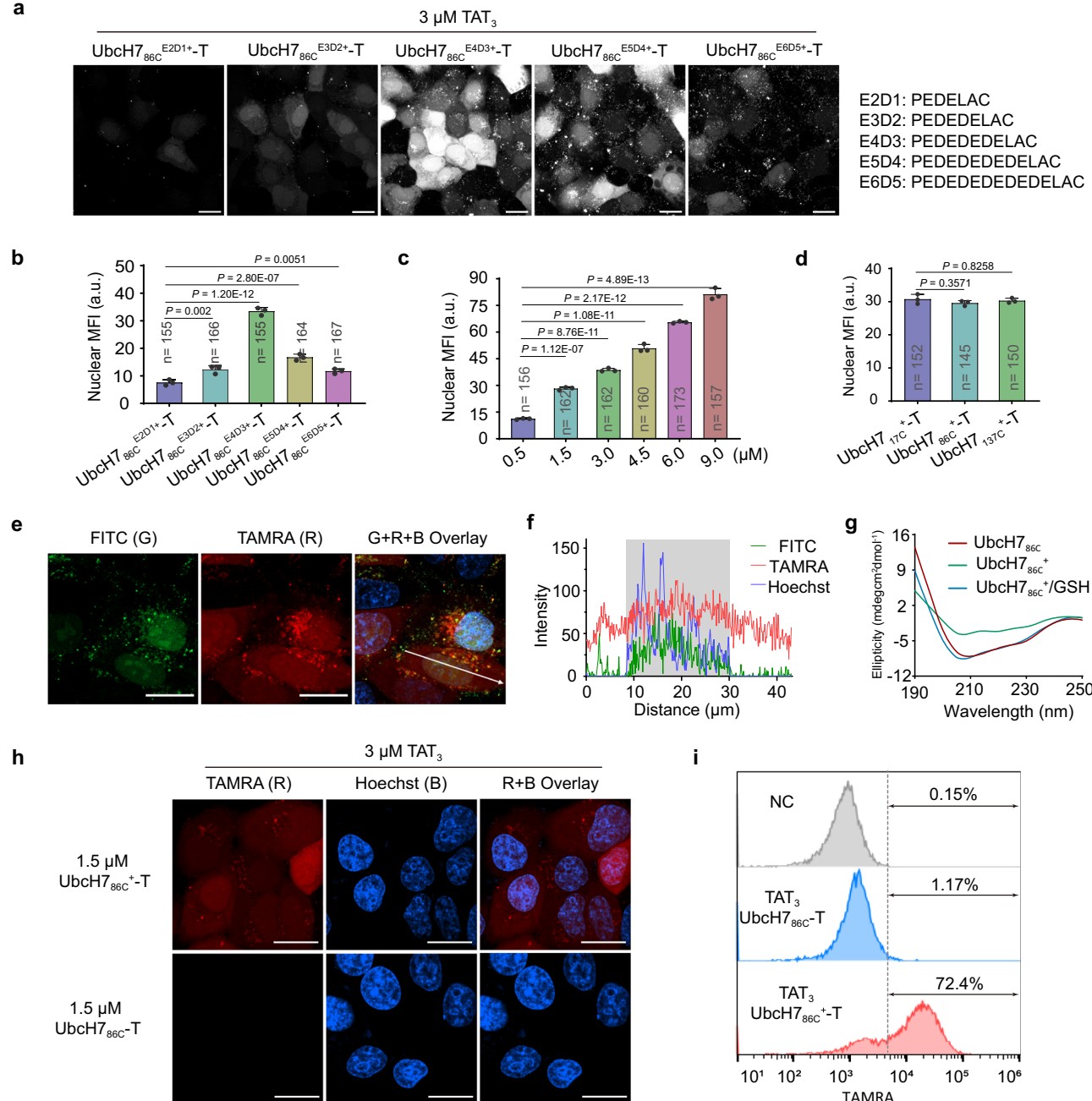

**Fig. 3 | Directed screening of anionic peptides for the cytosolic delivery of chemically-modified UbcH7₈₆C.** **a** Representative images of HepG2 cells treated with 1.5 μM five anionic peptide-modified UbcH7₈₆C-T, respectively, in the presence of 3 μM TAT₃ for 30 minutes at 37 °C. Scale bars, 20 μm. **b** Quantification of the nuclear MFI of the cells in Fig. 3a. The data are presented as the mean ± standard deviation. *P*-values were calculated via one-way ANOVA with Tukey's test correction. **c** Quantification analysis of nuclear TAMRA mean fluorescence intensity (MFI) of cells in Supplementary Fig. 6a. Data are presented as mean ± standard deviation. *P*-values were calculated by one-way ANOVA with Tukey's test correction. **d** Quantification of the nuclear MFI of the cells in Supplementary Fig. 6c. The data are presented as the mean ± standard deviation. *P*-values were calculated via one-way ANOVA with Tukey's test correction. **e** Images of HepG2 cells treated with

1.5 μM NLS-UbcH7₈₆C^(T+)-FITC and 3 μM TAT₃ for 30 minutes at 37 °C. The images shown are representative of independent biological replicates (*n* = 3). Scale bars, 20 μm. **f** Quantification of the fluorescence intensity along the white line shown in **e**. The gray box denotes the nuclear region. **g** CD spectra of UbcH7₈₆C, UbcH7₈₆C^+ and GSH-preincubated UbcH7₈₆C^+. **h** Confocal microscopy images of HepG2 cells treated with 1.5 μM UbcH7₈₆C^+-T or UbcH7₈₆C-T in the presence of TAT₃ for 30 minutes at 37 °C. The images shown are representative of independent biological replicates (*n* = 3). Scale bars, 20 μm. **i** Flow cytometry analysis of the HepG2 cells shown in Fig. 3h via detection of the MFI. "B", "R" and "G" stand for Hoechst (blue), TAMRA (red) and FITC (green) signals, respectively. Source data are provided as a Source Data file.

comparable intracellular delivery with UbcH7₈₆C^+-T in HepG2 cells (Fig. 3d and Supplementary Fig. 6c), indicating that the surface modification site does not significantly affect the delivery efficiency. Notably, serum-containing conditions were found to decrease delivery efficiency (Supplementary Fig. 6d), consistent with previous reports[11].

To verify the dissociation of E4D3 from UbcH7₈₆C in the cell, we labeled E4D3 with TAMRA (designated T + ) (Supplementary Fig. 1J), and conjugated it via disulfide bond to nuclear-localized UbcH7₈₆C (NLS-UbcH7₈₆C-FITC). After co-incubation of the adduct (NLS-UbcH7₈₆C^(T+)-FITC) and TAT₃ with HepG2 cells, CLSM images revealed

that the green fluorescence of NLS-UbcH7$_{86C}$ was predominantly localized in the nucleus, whereas the red fluorescence of TAMRA-E4D3 was diffusely distributed in both the cytosol and the nucleus (Fig. 3e, f). This spatial separation demonstrates that the disulfide-linked NLS-UbcH7$_{86C}^{T+}$-FITC undergoes reductive cleavage in the cytosol[21], releasing free NLS-UbcH7$_{86C}$-FITC and E4D3-TAMRA. Complementarily, the circular dichroism (CD) spectrum of 1 mM glutathione (GSH)-pretreated UbcH7$_{86C}^+$ showed very similar absorption to native UbcH7$_{86C}$ (Fig. 3g), indicating the reversion to an identical secondary structure.

Finally, we quantified the delivery efficiency by flow cytometry under optimized conditions (3 μM TAT$_3$ and 1.5 μM UbcH7$_{86C}^+$-T), and found that approximately 72.4% of the cells exhibited fluorescence signals (Fig. 3h-i). The HepG2 cells were then lysed and analyzed by fluorescence SDS-PAGE, which revealed the presence of UbcH7$_{86C}^+$-T in the cell lysate (Supplementary Fig. 7), confirming that UbcH7 was successfully delivered into living cells with minimal cytotoxicity by anionic surface modification.

### UbcH7$_{86C}^+$ is internalized via macropinocytosis

To probe the interaction between UbcH7$_{86C}^+$-T and TAT$_3$, we performed native gel electrophoresis analysis[12]. As showed in Fig. 4a, co-incubation of UbcH7$_{86C}^+$-T with TAT$_3$ generated a band shift above the original UbcH7$_{86C}^+$-T. This shifted band exhibited concentration-dependent attenuation with increasing heparin concentrations, accompanied by a concomitant intensification of the original band. The complete disappearance of the band shift occurred when the concentration of heparin exceeded 2 μM, demonstrating that the non-covalent complex formation between UbcH7$_{86C}^+$-T and TAT$_3$ is mediated by electrostatic interactions[14].

We next investigated the delivery mechanisms of UbcH7$_{86C}^+$-T and TAT$_3$, starting with 70 kDa dextran-FITC, a well-established marker of macropinocytosis. HepG2 cells co-incubated with dextran-FITC and TAT$_3$ showed diffuse green fluorescence in the cytoplasm (Fig. 4b and Supplementary Fig. 8a) as reported in previous work[13,22]. Further experiments co-incubating cells with dextran-Cy5, UbcH7$_{86C}^+$-T and TAT$_3$ showed simultaneous red fluorescence of UbcH7$_{86C}^+$-T and purple fluorescence of dextran-Cy5 in the cells (Fig. 4c and Supplementary Fig. 8b), indicating that UbcH7$_{86C}^+$-T was delivered into the cells via the macropinocytosis in the same way as 70 kDa dextran. Co-localization analysis using the LysoTracker showed that less than 40% of UbcH7$_{86C}^+$-T signals co-localized[23] with the LysoTracker (Fig. 4d), suggesting that the majority of UbcH7$_{86C}^+$-T escapes from endosomes[24].

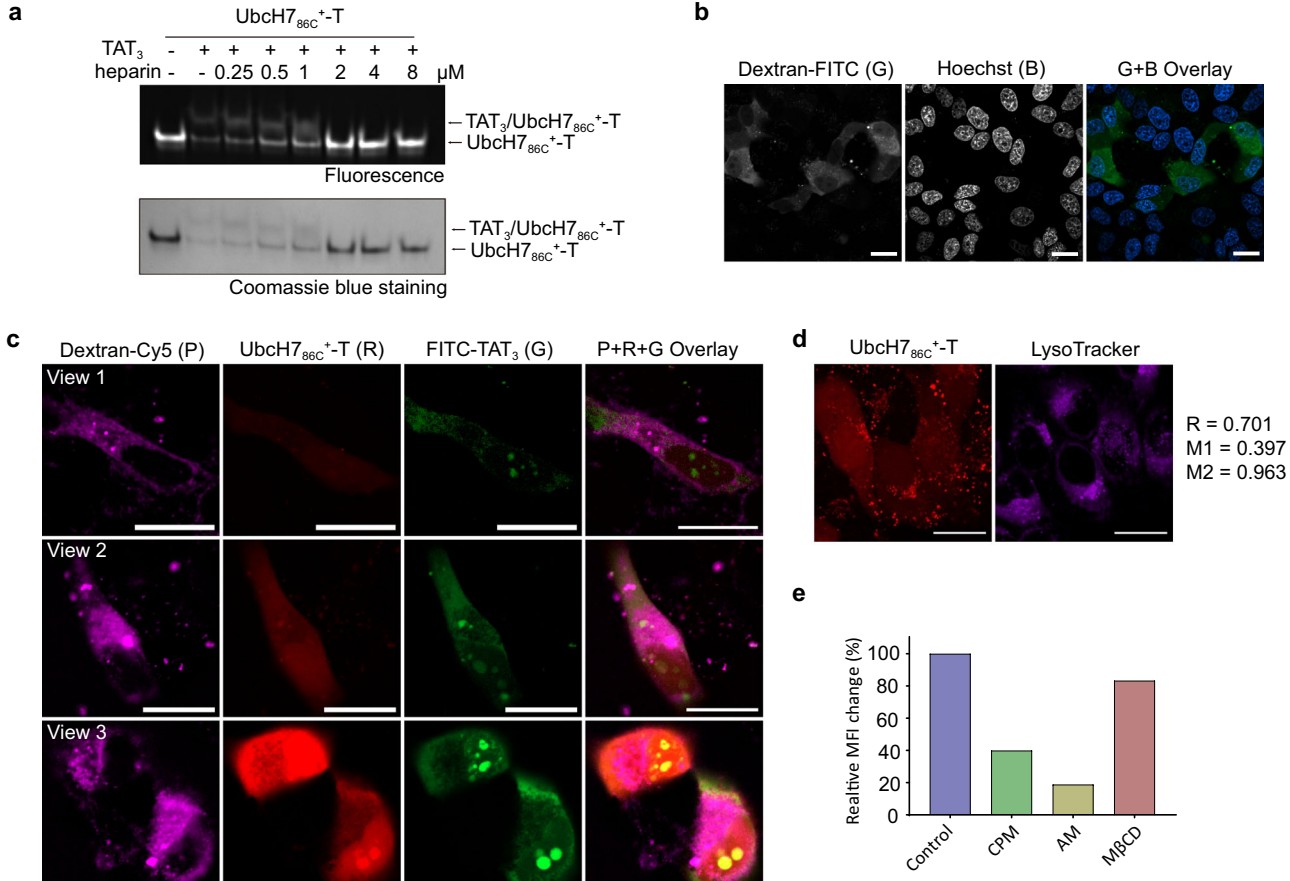

**Fig. 4 | Study of the internalization mechanism. a** Coomassie blue staining and fluorescence images of the samples of UbcH7$_{86C}^+$-T and TAT$_3$ in the presence of different concentrations of heparin. The images shown are representative of independent biological replicates ($n$ = 2). **b** Fluorescence images of HepG2 cells treated with 1 mg/mL 70 kDa dextran-FITC and 3 μM TAT$_3$ at 37 °C for 30 minutes. The images shown are representative of independent biological replicates ($n$ = 3). **c** Representative images of HepG2 cells treated with 1 mg/mL 70 kDa dextran-Cy5, 1.5 μM UbcH7$_{86C}^+$-T and 3 μM FITC-TAT$_3$ simultaneously at 37 °C for 30 minutes. The images shown are representative of independent biological replicates ($n$ = 3). **d** Whole cell co-localization analysis of UbcH7$_{86C}^+$-T and LysoTracker signal. Co-localization images from the left: red channel (UbcH7$_{86C}^+$-T); purple channel (LysoTracker). The images shown are representative of independent biological replicates ($n$ = 3). **e** Changes in the relative MFI of HepG2 cells pretreated with various endocytosis inhibitors followed by incubation with UbcH7$_{86C}^+$-T and TAT$_3$ for 30 minutes. "B", "R", "G" and "P" stand for Hoechst (blue), TAMRA (red), FITC (green) and Cy5 (purple) signals, respectively. Scale bars, 20 μm. Source data are provided as a Source Data file.

We also explored the role of transduction in the cellular uptake of UbcH7$_{86C}^+$-T. Unlike macropinocytosis, transduction can occur at 4 °C[25,26] and it has been reported that the accumulation of CPPs at the cell membrane by transduction would result in a local membrane flip-flop[27]. Our observations showed no detectable red fluorescence in cells co-incubated with UbcH7$_{86C}^+$-T and TAT$_3$ at 4 °C (Supplementary Fig. 8c). Furthermore, using annexin V, a specific binding protein for membrane phosphatidylserine, we did not observe any enrichment of phosphatidylserine during UbcH7$_{86C}^+$-T internalization (Supplementary Fig. 8d). These results suggest that transduction is not the primary pathway for the delivery of UbcH7$_{86C}^+$-T and TAT$_3$.

Finally, we quantified the delivery efficiency of UbcH7$_{86C}^+$-T in the presence of various pathway inhibitors[28] by flow cytometry. Inhibitors, including chlorpromazine (CPM, an inhibitor of clathrin-mediated endocytosis), amiloride (AM, an inhibitor of macropinocytosis), methyl-beta-dextrin (MβCD, a cholesterol depleting substance), were used as described in previous studies[29–32]. As shown in Fig. 4e and Supplementary Fig. 8e, the most pronounced fluorescence attenuation (reaching only 20% of the control group) was observed under AM treatment. Taken together, these results demonstrate that the E4D3-modified UbcH7$_{86C}$ can form electrostatic complexes with TAT$_3$, and then be internalized via macropinocytosis and reach the cytosol by escaping from endosomes.

## Synergizing anionic surface modification and electrostatic interaction with TAT$_3$ enables cytosolic delivery of various cargoes into living cells

We then attempted to deliver a broad palette of proteins with different molecular weights (MWs) and isoelectric points (pIs) into cells using the anionic surface modification strategy (Fig. 5a). For the preparation of E4D3-modified protein cargoes, proteins containing free cysteine residues (e.g., NLS-UbcH7, BFP, RFP, Lifeact, β-Gal, and BSA) were directly conjugated to DTNB-activated E4D3 via disulfide bond as described above. For proteins lacking free cysteines (including RNase A, IgG, HRP, and Ub), we implemented a thiolation protocol using a widely adopted reagent 2-iminothiolane[33] (25 equiv.) to generate free thiol, followed by E4D3 conjugation via disulfide bond to prepare E4D3-modified adduct (cargo$^+$) (Fig. 5b). Mass spectrometric analysis confirmed the successful introduction of up to four free thiol groups per protein molecule after 2-iminothiolane treatment, and finally installed one to two E4D3 per protein molecule (Supplementary Fig. 9a, 9b).

We first evaluated the cellular delivery of the positively charged protein NLS-UbcH7 (pI: 9.4, MW: 19.6 kDa). Co-incubation of NLS-UbcH7$^+$-FITC and TAT$_3$ resulted in distinct nuclear green fluorescence in treated cells (Fig. 5c). Quantitative analysis showed that approximately 86% of the FITC signals co-localized with Hoechst[34], and 85% of the Hoechst signal colocalized with FITC (Supplementary Fig. 9c). In stark contrast, non-modified NLS-UbcH7-FITC was not delivered into cells when co-incubated with TAT$_3$ (Supplementary Fig. 10a). We then investigated the delivery of neutrally charged red fluorescent protein (RFP, pI: 7.3, MW: 26.9 kDa), which was very difficult to deliver intracellularly when co-incubation with tri-cTat in previous studies[14]. Both confocal microscopy and Z-stack imaging revealed diffuse red fluorescence in HepG2 cells co-incubated with RFP$^+$ and TAT$_3$, whereas control groups (cells incubated with RFP and TAT$_3$) showed minimal fluorescence (Fig. 5c, Supplementary Fig. 9d and 10b). Further experiments extended to proteins with lower pIs including blue fluorescent protein (BFP, pI: 6.6) and bovine serum albumin (BSA, pI: 4.7, MW: 69.3 kDa), only E4D3-modified adducts showed successful intracellular delivery when co-incubated with TAT$_3$ (Fig. 5c, Supplementary Fig. 10c, 10d).

We next investigated intracellular delivery of the Lifeact peptide (pI: 4.9, MW: 1.9 kDa), which can target filamentous actin (F-actin) in living cells[35]. CLSM revealed distinct green fluorescence in cells

co-incubated with Lifeact$^+$-FITC and TAT$_3$ (Fig. 5c and Supplementary Fig. 10e). Quantitative analysis showed a high degree of colocalization between Lifeact$^+$-FITC signals and the F-actin marker SiR700-Actin (M1 = 0.589, M2 = 0.528) (Supplementary Fig. 9e). Moreover, after quantified by image-based analytical method[14], we found that all fluorescence labeled proteins exhibited transduction efficiencies of over 55% (Fig. 5d). We also examined the cellular delivery of two enzyme cargoes: ribonuclease A (RNase A, pI: 8.6, MW: 13.7 kDa)– known to induce cell death[36], and horseradish peroxidase (HRP, pI: 7.0, MW: 40 kDa)–capable of catalyzing the conversion of the colorless substrate tetramethylbenzidine (TMB) to a blue product in the presence of hydrogen peroxide[37]. HepG2 cells treated with RNase A (up to 3 μM), TAT$_3$ alone, or their mixture maintained ≥90% cell viability. In contrast, co-incubation with RNase A$^+$ and TAT$_3$ resulted in concentration-dependent cell death (Fig. 5e). In parallel, functional assessment of HRP delivery showed that HepG2 cells co-incubated with TAT$_3$ and HRP$^+$ exhibited the most intense blue color in TMB assays (Fig. 5f). These functional assays confirm successful intracellular delivery of biologically active RNase A$^+$ and HRP$^+$ into target cells.

Finally, we explored the molecular weight capacity of our strategy for protein delivery, starting with full-length immunoglobulin G antibody (IgG, MW: 150 kDa). CLSM showed diffuse green fluorescence in cells incubated with IgG$^+$-FITC and TAT$_3$ (Fig. 5c and Supplementary Fig. 10f), indicating successful intracellular delivery of IgG$^+$-FITC. β-Gal (MW: 430 kDa), a tetrameric complex enzyme capable of catalyzing the hydrolysis of colorless 5-bromo-4-chloro-3-indolyl-β-D-galactoside (X-Gal) to form blue products[38], was then tested. HepG2 cells incubated with β-Gal$^+$ and TAT$_3$ showed blue staining after treatment with the X-Gal kit (Fig. 5g). As a control, direct conjugation of β-Gal with R$_{10}$ resulted in significantly weaker intracellular blue signals under the same conditions (Fig. 5g). These results demonstrate the applicability of the strategy of synergizing anionic surface modification and electrostatic interaction for the cytosolic delivery of different cargoes in the bioactive form.

## Cytosolic delivery of proteins in different mammalian cell lines and plants

Next, we investigated the efficacy of the strategy in different cell lines. Co-incubation of UbcH7$_{86C}^+$-T and TAT$_3$ in BEAS-2B (human bronchial epithelium), HeLa (human cervical adenocarcinoma) and Jurkat (human T lymphocyte) cells all resulted in diffuse red fluorescence similar to HepG2 cells (Fig. 6a and Supplementary Fig. 11a), demonstrating the applicability of the strategy in multiple mammalian cells.

In contrast to mammalian cells, plant cells possess a structurally dense cell wall outside the plasma membrane (Fig. 6b), which imposes additional constraints on the cellular delivery of exogenous biomolecules[39]. Herein, Arabidopsis thaliana (A. thaliana) and Nicotiana tabacum (N. tabacum) were chosen as model plants. Leaves were infiltrated[40] with a mixture of UbcH7$_{86C}^+$-T and TAT$_3$ and incubated for 1 hour, then the leaves were re-infiltrated in situ with QSY21-NHS (N-hydroxysuccinimide linked fluorescence quencher moiety)[41] to quench the extracellular fluorescence of TAMRA. The plant leaves were then cut and washed with water to remove protein samples adsorbed on the leaf surface and subjected to CLSM. As shown in Fig. 6c, obvious red fluorescence was observed in the epidermal cells and was distributed around the cell contour, similar to previous work[42]. In contrast, leaves co-incubated with UbcH7$_{86C}$-T and TAT$_3$ showed negligible red fluorescence under identical conditions. Furthermore, we tested the delivery of the negatively charged protein Ub. Distinct red fluorescence was only observed in plant leaves when co-incubated with Ub$^+$-T and TAT$_3$ (Fig. 6c). These results demonstrate the successful delivery of both positively charged (UbcH7$_{86C}$) and negatively charged (Ub) proteins to A. thaliana and N. tabacum leaves using the strategy of synergizing anionic surface modification and

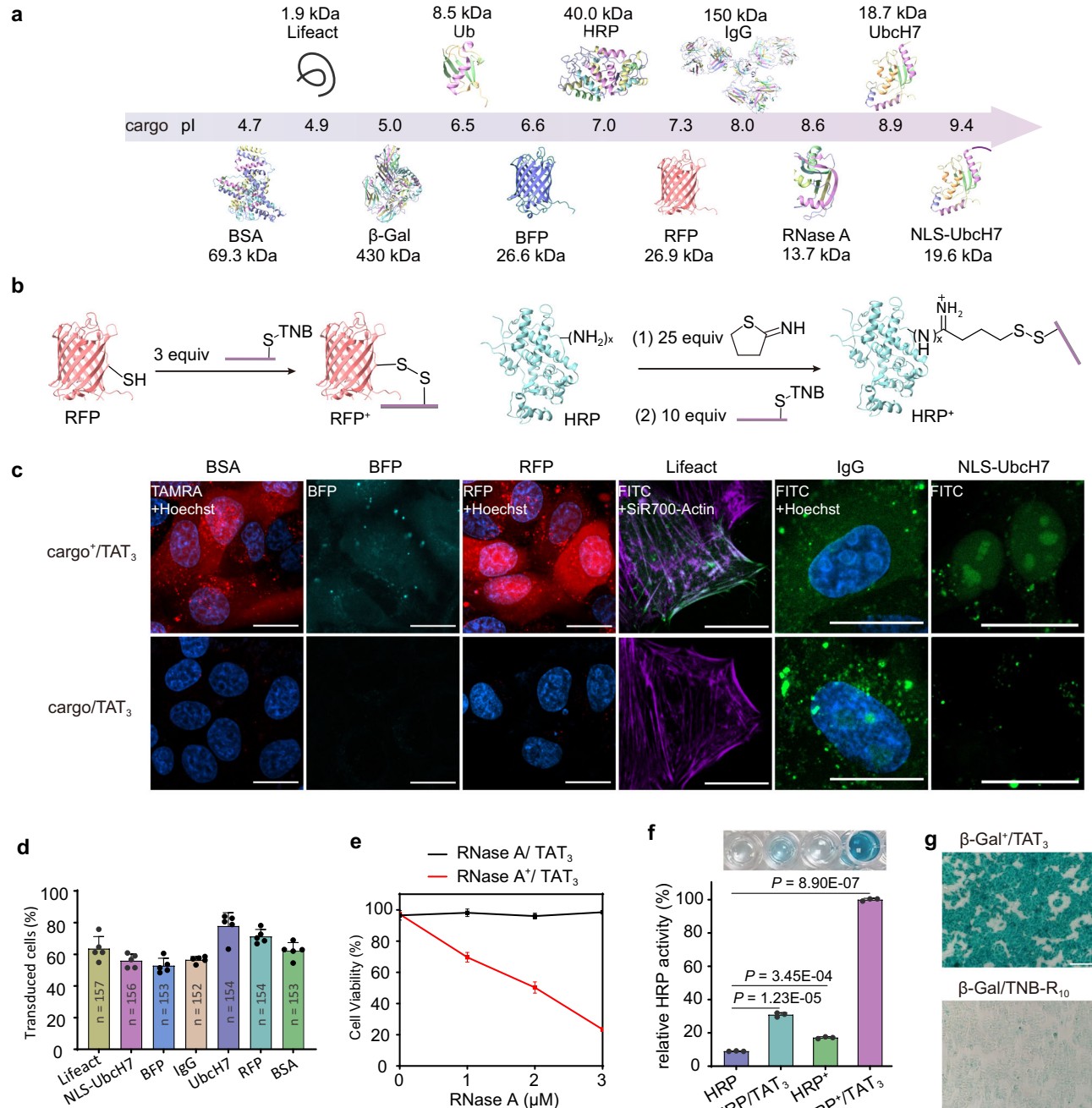

**Fig. 5 | Synergizing anionic surface modification and electrostatic interactions for the cytosolic delivery of various cargoes. a** Proteins and peptides with different molecular weights and isoelectric points used in this study. **b** Scheme showing the anionic surface modification of protein cargoes with or without free cysteines. **c** Representative images of HepG2 cells incubated with anionic modified protein cargoes and TAT₃. Scale bars, 20 μm. **d** Quantification of percentage of the transduced cell for various cargoes. The results are the average of five biological replicates and presented as the mean ± standard deviation. **e** Concentration-dependent cytotoxicity of RNase A⁺ and TAT₃. The results are the average of three biological replicates and presented as the mean ± standard deviation. **f** HRP enzyme activity in treated cells stained with TMB. HRP and HRP⁺ alone were used as controls. The results are the average of three biological replicates and presented as the mean ± standard deviation. *P*-values were calculated by one-way ANOVA with Tukey's test correction. **g** X-Gal staining of cells treated with β-Gal⁺ in the presence of TAT₃ (top row) or TNB-R₁₀ (bottom row). Scale bars, 200 μm. Source data are provided as a Source Data file.

electrostatic interaction. In addition, we evaluated the delivery efficiency of disulfide-linked cR₁₀-conjugated Ub-TAMRA (cR₁₀-Ub-T), but observed only weak fluorescence signals, no enhanced fluorescence was detected even when TNB-R₁₀-ILFF[43] was added (Supplementary Fig. 11b). Taken together, these data confirm the broad applicability of our strategy for the delivery of protein cargoes to mammalian cell lines and plant cells.

## Chemical synthesized surface modified E2-Ub probes empowers the exploration of E2-E3 interactome in living cells

Encouraged by the successful delivery of UbcH7₈₆C into various cell lines, we next attempted to deliver E2-Ub probes to profile their partner E3s in living cells, with the goal of mapping the E2-E3 network in physiological processes[44]. Inspired by the biotinylated E2-Ub photocrosslinking probe (UBE2D3-Ub Bpa) reported by Mathur et al.[45], we

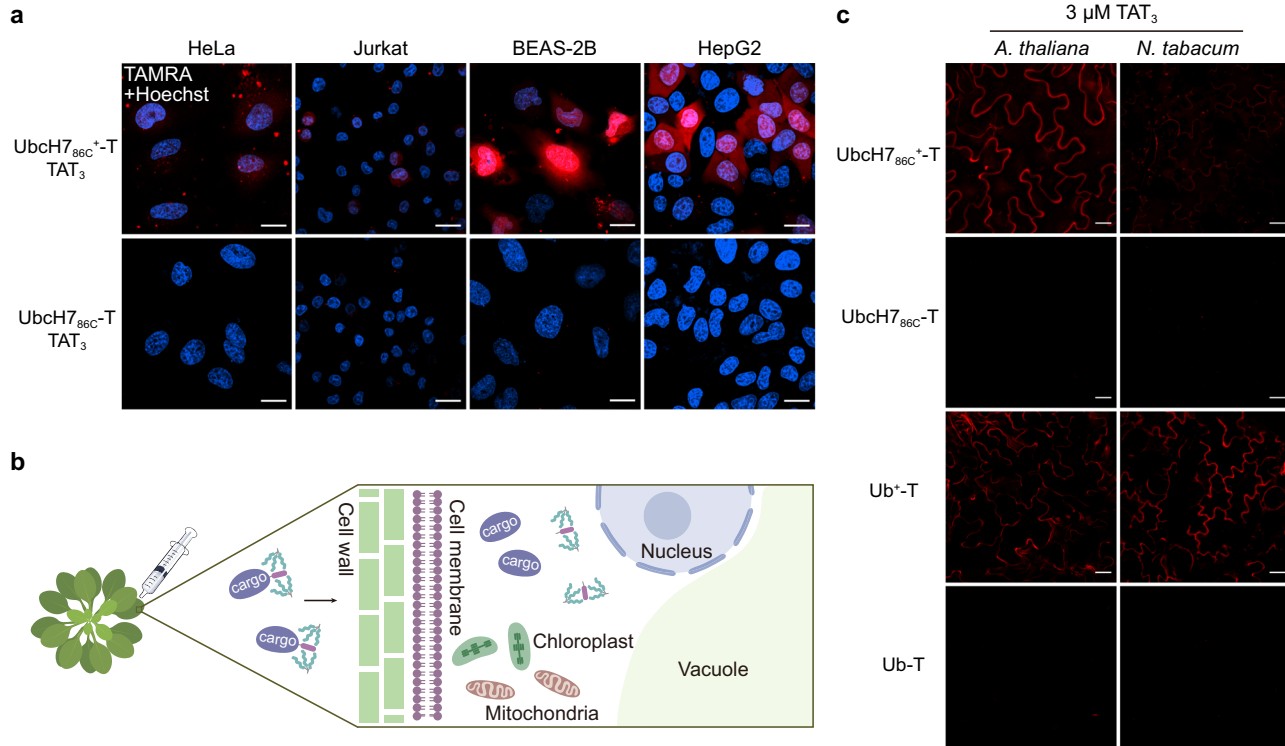

**Fig. 6 | Delivery of protein cargoes in different mammalian cell lines and plants.** **a** Confocal microscopy images of four mammalian cell lines treated with 1.5 μM UbcH7$_{86C}^{+}$-T or UbcH7$_{86C}$-T in the presence of TAT$_3$ for 30 minutes at 37 °C. The images shown are representative of independent biological replicates ($n = 3$). **b** Scheme showing the architecture of plant cells and the workflow of plant infiltration. **c** Representative images of protein cargoes delivery into *A. thaliana* and *N. tabacum* leaves after treatment with QSY21-NHS. The images shown are representative of independent biological replicates ($n = 3$). Scale bars, 20 μm.

modified its surface with E3D4 and applied it to capture RING E3s (the largest E3 subfamily) in living cells. For this end, the E2 mutant (UBE2D3 C85K S22R) was recombinantly expressed in which the catalytic cysteine was mutated to lysine to allow the formation of an isopeptide bond with ubiquitin (Ub), an isopeptide bond that was demonstrated by structural analysis to be an acceptable structural mimetic of the native thioester[46]. Moreover, serine was replaced by arginine to disrupt the self-association of the probe. The biotinylated Ub Bpa31 (substitution of benzophenone (Bpa) at position 31 of ubiquitin), obtained by Fmoc-based solid-phase peptide synthesis (SPPS)[47–49], was then conjugated to the E2 mutant with an E1-activating enzyme to form isopeptide bonds[50]. Finally, the E4D3 peptide was conjugated via disulfide bonds and fluorescently labeled with NHS-TAMRA to form the UBE2D3-Ub Bpa probe$^{+}$-T (Fig. 7a). The CLSM images revealed diffuse red fluorescence in HeLa cells with small number of puncta after incubation with probe$^{+}$-T and TAT$_3$ (Fig. 7b), which is similar to the previous studies on delivering large-molecular-weight proteins such as Fab fragments[14].

We next investigated the ability of probe$^{+}$ to profile endogenous RING E3 in a proteome-wide manner in response to epidermal growth factor (EGF)[51] stimulation. HeLa cells were first preincubated with MG132 and bafilomycin for 6 hours and then stimulated with EGF. The experimental cells (A) were then incubated with probe$^{+}$ and TAT$_3$, while the control cells (B) were incubated with probe$^{+}$ alone, then both cells were resuspended in cold PBS and irradiated with 365 nm UV on ice to activate Bpa[45]. Finally, the cell lysates were collected, and biotinylated proteins were enriched using streptavidin magnetic beads for LC-tandem MS analysis (Fig. 7c). Among the total proteins identified by MS, 42 were RING E3s, 20 of which were UBE2D3 partners (Supplementary Table 1). We then filtered the detected E3s using criteria (fold change > 2 and *P*-value < 0.05) to ensure that the detected E3 signals

correlated with E3 activity or abundance. Our results showed that 21 RING E3s were significantly enriched, and 11 of these E3s have been reported to catalyze protein ubiquitination in cooperation with UBE2D3. Furthermore, another 7 RING E3s were reported to catalyze protein ubiquitination in cooperation with other E2s belonging to the UBE2D family (Fig. 7d and Supplementary Table 2). Notably, the UBE2D3-Ub Bpa probe was previously used to profile RING E3s in EGF-stimulated HEK293T cell lysates[45]. A comparison of these two data sets revealed that 10 partner E3s were labeled in both living cells and lysates. We identified additional E3s, such as UHRF1, TRIM32, ZNF598, RNF168 and RNF181, and Mathur et al.[45] also reported different E3s, including BRCA1, BRE1A, BRE1B, HLTF and PCGF6. Together, these findings collectively show the complementarity between the two approaches, offering valuable tools for studying E2–E3 pairing relationships in physiological or pathological processes.

## Discussion

In recent years, CPP-mediated delivery has emerged as a prevalent method to deliver biomolecules into cells, but a general platform that can deliver various proteins at low micromolar concentrations remains inaccessible. Herein, we reported a strategy for efficient cytosolic protein delivery by engineering cargoes with anionic peptide patch E4D3 and using trimeric peptide clusters as carriers. In this context, conjugation of E4D3 to protein cargoes creates a localized negative surface charge, which enhances electrostatic interaction with TAT$_3$. This ultimately enables the efficient delivery of a broad range of proteins. The anionic peptide E4D3 can be efficiently conjugated to protein cargoes via a disulfide bond under mild conditions, and the whole process is easy to implement and cost-effective. Proteins without free cysteines can be thiolated to allow disulfide bond exchange reactions. In these cases, thiolation may affect the function of certain proteins,

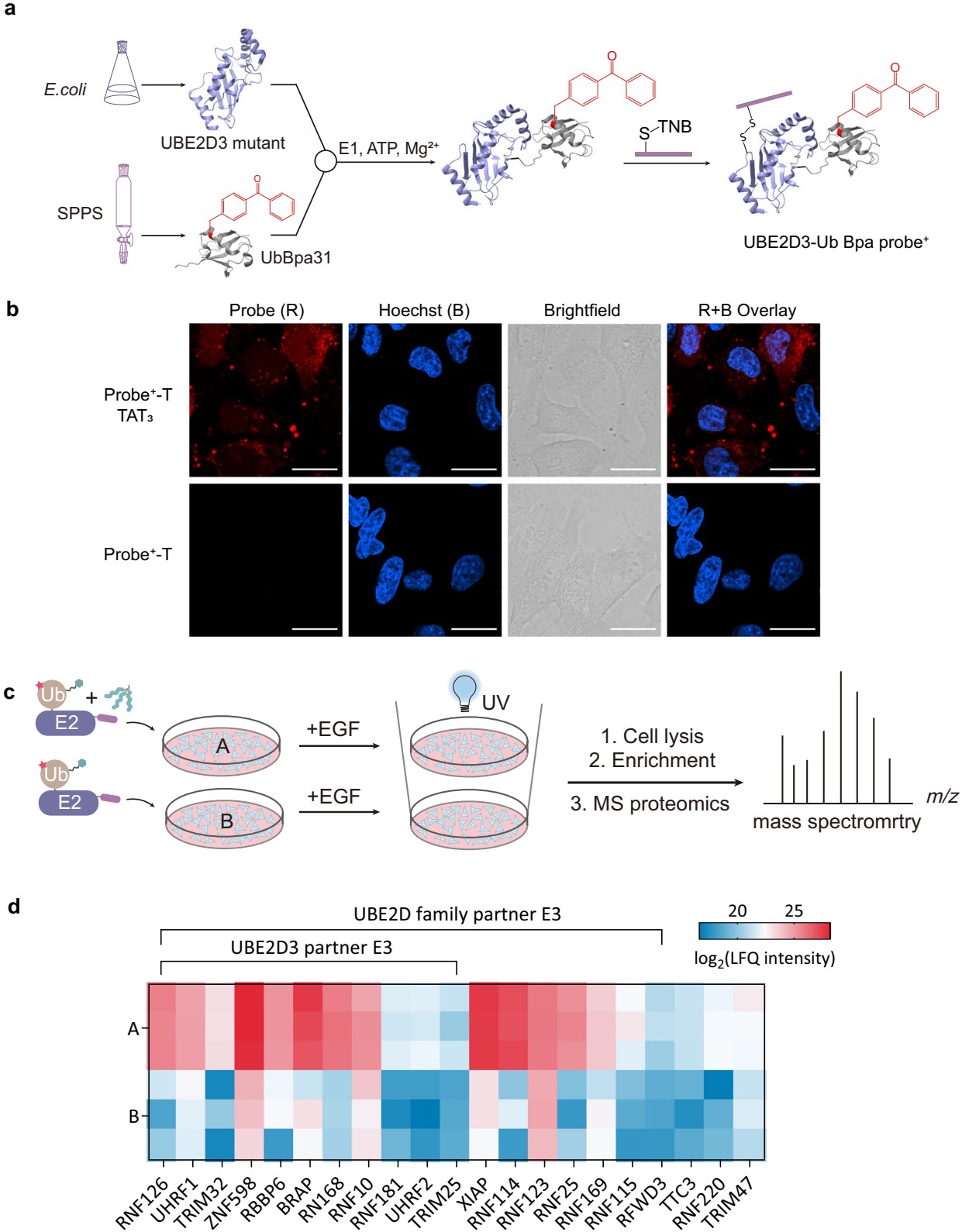

**Fig. 7 | Cytosolic delivery of E2-Ub photocrosslinking probes to HeLa cells.**
**a** Synthetic scheme of the E2-Ub photocrosslinking probes. **b** Representative images of probe⁺-T in HeLa cells. "B" and "R" stand for Hoechst (blue) and TAMRA (red) signals, respectively. The experiment was repeated three times. Scale bars, 20 μm. **c** Schematic depicting the proteomic workflow with biotinylated E2-Ub photocrosslinking probes⁺. **d** The significantly enriched RING E3s and their pairing relationships with UBE2D3. Only proteins met the criteria of *P*-value < 0.05 (Ben-jamini-Hochberg correction for multiple comparisons) and fold change > 2 were considered significantly enriched. Statistical significance was determined by a two-sided *t*-test. Heatmap of log₂-transformed LFQ intensities for UBE2D family partner E3 ligases. Source data are provided as a Source Data file.

therefore trace-free protein modification methods are still necessary. Upon entry into the cytoplasm, the disulfide-linked E4D3 peptide automatically dissociates to release native proteins. In addition to delivering proteins into mammalian cells, this strategy also works in model plants, including *A. thaliana* and *N. tabacum*. Finally, by intracellularly delivering a synthetic E2-Ub probe into HeLa cells, we have physiologically profiled cellular E2-E3 pairing networks.

Based on these results, our study showed that both the CPP and the anionic peptide are key factors for successful delivery. Compared to tri-cTat, $TAT_3$ is more suitable for our protocol, probably due to its more concentrated positive charge distribution, which enhances electrostatic interaction with anionic peptides. Direct screening revealed that E4D3 modification achieves higher delivery efficiencies than four other four peptides tested. This optimal performance may be due to the well-balanced negative charge density of E4D3 - sufficient to maintain electrostatic interactions with $TAT_3$ while avoiding excessive charge shielding effects, and this charge engineering approach echoes the efforts in earlier studies[52]. It is worth mentioning that some negatively charged cargoes, such as BSA and BFP, can only be delivered into cells after E4D3 modification. We speculate that this is because the average charge distribution of the entire protein may be insufficient to form a high-density local negative charge "patches" capable of generating electrostatic interactions with $TAT_3$, whereas E4D3 modification significantly increases the local negative charge density of the cargo protein, thereby generating a stronger charge interaction with $TAT_3$.

Our strategy demonstrates the capacity for intracellular delivery of macromolecular proteins, including IgG (150 kDa) and β-galactosidase (430 kDa), and works in plants with cell walls (*A. thaliana* and *N. tabacum*). The delivery efficiency is superior to covalent conjugation with CPPs (e.g., $R_{10}$ and $TAT_3$) at equivalent concentrations (Supplementary Fig. 11b and 12), suggesting that a higher density of CPPs may enhance cellular delivery efficiency. By combining protein chemical synthesis, our strategy enables the effective cytosolic delivery of both recombinantly expressed proteins and chemically-customized protein tools, positioning it as a practical technology for cellular protein applications. Our future work will explore the use of this strategy to deliver more types of synthetic protein probes and therapeutically relevant proteins[53–55] into cells, while exploring its application potential in complex biological systems beyond cultured cells, including tissue culture and even model organisms.

## Methods
### General materials and methods
Fmoc-amino acids were purchased from GL Biochem (China). Resins were purchased from Tianjin Nankai HECHENG S&T Co., Ltd (China). Coupling reagents were purchased from CSBio Ltd. (China). Dichloromethane (DCM) and N, N-Dimethylformamide (DMF) were purchased from J&K Scientific Ltd (China). Ni-NTA resin was purchased from LABLEAD (China). Chemically competent *BL21* (DE3) cells were purchased from TransGen Biotech ltd (China). Inorganic salts were purchased from Sinopharm Chemical Reagent Co., Ltd (Shanghai). Yeast extract and Tryptone were purchased from Sigma-Aldrich. DMEM, RPMI-1640, PBS and penicillin/streptomycin were purchased from Gibco. FBS was purchased from Lonsera. 35 mm glass-bottom dishes were purchased from Biosharp. Metabolic inhibitors and fluorescent molecules were purchased from MedChem-Expresss (USA). Living cell dyes such as Hoechst 33258, LysoTracker™ Deep Red were purchased from Thermo Fisher Scientific (USA), and SiR700-Actin were purchased from Spirochrome (Switzerland). TMB, X-Gal staining kit, CCK8 kit and Calcein/PI cytotoxicity assay kit were purchased from Beyotime (China). RNase A (Ribonuclease A) was purchased from Yeasen Biotechnology Co., Ltd. (China). Goat anti-Rabbit IgG (Immunoglobulin G) and BSA

(bovine serum albumin) was purchased from Sangon Biotech (China). HRP (Horseradish Peroxidase) was purchased from Aladdin (China). β-Gal (β-Galactosidase) was purchased from Solarbio (China). Peptide were analyzed or purified with RP-HPLC (SHI-MADZU, Prominence LC 20AT). For analysis, analytical Welch XB-C18 columns (4.6 × 250 mm, 5 μm, 120 Å, 1.0 mL/min) were used. For purification, semi-preparative Welch XB-C18 (250 × 10 mm, 5 μm, 120 Å, flow rate 4.0 mL/min) were used. Analysis and purification were monitored at 214 and 254 nm. Solvents were sonicated for 30 min before use. Analysis and purification condition: a linear gradient of 5–80% acetonitrile (with 0.08% v/v TFA) in water (with 0.1% v/v TFA) over 30 min. Fast Protein Liquid Chromatography (FPLC) was performed on an AKTA purifier (GE Healthcare Life Science) with column of Superdex TM 75 Increase 10/300 GL. Phosphate buffer (PBS, pH 7.4) was filtered with 0.22 μm filter paper, sonicated for 30 min and precooled for 10 min before use. Purification was monitored at 280 nm wavelength. ESI-MS were measured on a Shimadzu LC/MS-2020 system. High resolution mass spectra were measured on a Q Exactive Plus mass spectrometer coupled to an Easy-nLC1200 system running on 50% acetonitrile with 0.1% formic acid using the Xcalibur software. Protein spectra were devonvoluted using the UniDec tool. Automated Fmoc-based microwave peptide synthesis was performed on a Liberty Blue 2.0™ automated microwave peptide synthesizer (CEM Corporation, North Carolina, USA).

### Experimental model
For bacteria cultures, all recombinant expressed protein genes were synthesized by Genscript (Nanjing) and cloned into pET-28a vector. Protein (RFP, BFP, UbcH7 mutants, and UBE2D3 C85KS22R) plasmids were transfected into *E. coli* BL21 (DE3) cells and cultured in Luria-Bertani (LB) medium at 18 °C. For mammalian cell cultures, HepG2(CL-0103), HeLa (CL-0101) and Jurkat (CL-0129) cells were purchased from Procell, BEAS-2B was a generous gift from Guiran Xiao Lab (Hefei University of Technology, Hefei). HepG2, HeLa and BEAS-2B cells were cultured in DMEM supplemented with 10% fetal bovine serum (FBS) and 100 units/mL penicillin and 0.1 mg/mL streptomycin at 37 °C in an atmosphere of 5% $CO_2$, while Jurkat cells were cultured in RPMI-1640 supplemented with 10% FBS and penicillin/streptomycin at 37 °C in an atmosphere of 5% $CO_2$. For plant cultures, take several *Nicotiana tabacum* or *Arabidopsis thaliana* seeds and spread them evenly over the surface of the soil. Cover the soil with cling film to keep the soil moist and leave a few small holes for gas exchange with the outside. After one week of incubation at a constant temperature and humidity, the seedlings were transferred to individual pots (one plant per pot) and the nutrient solution was replenished every three days. Plants for fluorescence imaging were grown for 4-5 weeks under a 16 hours light/ 14 hours dark regimen at 22 °C/20 °C.

### Protein expression and purification
Protein (RFP, BFP, UbcH7 mutants and UBE2D3 C85KS22R) plasmids were transfected into *E. coli* BL21 (DE3) cells. Cells was grown in LB containing 0.1 mg/mL Kanamycin at 37 °C to an OD600 of 0.8. IPTG was then added to a final concentration of 0.4 mM and the mixture was incubated for 12 hours at 16 °C. Cell precipitates were collected by centrifugation at 4 × g, 8 °C for 15 minutes, and then resuspended in lysis buffer (20 mM Tris, 150 mM NaCl, pH 7.4) and sonicated in an ice bath. The suspension was centrifuged at 4 °C, 10 × g for 30 minutes to obtain the supernatant. All proteins fused to the His6 tag were purified on Ni-NTA columns. Proteins were washed with wash buffer (20 mM Tris, 150 mM NaCl, 40 mM imidazole, pH 7.4) and eluted with elution buffer (20 mM Tris, 150 mM NaCl, 250 mM imidazole, pH 7.4). After concentration by 3 K MWCO tubes, the proteins were further purified on a Superdex 75 10/300 GL column. Purified proteins are used immediately for experiments or stored at −80 °C until use.

## Synthesis of peptides

All peptides used in this study were obtained via the standard Fluorenyl-methoxycarbonyl (Fmoc) based solid phase peptide synthesis (SPPS) on Rink amide resin (0.29 mmol/g) or Wang resin (0.44 mmol/g). Fmoc-based SPPS was performed using a Liberty Blue 2.0™ automated microwave peptide synthesizer (CEM Corporation, North Carolina, USA). Fmoc deprotection: The Fmoc group was removed using a 20% solution of piperidine in DMF at 90 °C (1 × 1 minute, 5 mL). The resins were then washed with DMF (Dimethyl-formamide, 4 × 5 mL). Amino acid couplings: The amino acids (10.0 equiv. to resin loading), oxyma (Ethyl cyanoglyoxylate-2-oxime, 10.0 equiv.) and DIC (N,N′-Diisopropylcarbodiimide, 10.0 equiv.) were dissolved in DMF, with amino acid final concentration at 0.2 mol/L. Afterward, the solution was added to the resins and the reaction for 2 minutes at 90 °C. The resins were then washed with DMF (2 × 5 mL). Resin cleavage: A cleavage cocktail (TFA / Phenol / Thioanisole / 1, 2 ethanedithiol / water / Triisopropylsilane, 83/5/5/2.5/2.5/2, v/v/v/v/v/v) was added to dried resin, after which the mixture was shaken at 37 °C for 3 hours. The resin was then filtered and the resulting cleavage solutions were volatilized under a stream of nitrogen. Cold ether was added to the residue and the mixture was thoroughly mixed and centrifuged. The supernatant was discarded to afford crude peptide.

$TAT_3$ peptides was synthesized by incorporation of lysine that was orthogonally protected with Mtt (methyltrityl) group. Specifically, Fmoc-Lys(Mtt)-OH was used in solid-phase peptide synthesis to give resins loaded with linear peptides. After completion of the linear peptide synthesis, the N-terminal amino blocked by blocking solution (Ac₂O/DIEA/DMF, 1/1/8, v/v/v) at 37 °C (2 × 5 minutes). Next, the Mtt group was removed using 2% TFA (v/v) with 1% TIPS (Triisopropylsilane, v/v) in DCM (6 × 5 minutes). Finally, an automated microwave peptide synthesizer was used to couple the TAT sequence onto the amino of the lysine side-chain. Fmoc was removed by 20% solution of piperidine in DMF at 90 °C (1 × 1 minutes), and then the resins were washed by DMF (4 × 10 mL). Amino acids (16.7 equiv. to resin loading), oxyma (16.7 equiv.) and DIC (16.7 equiv.) were dissolved in DMF and added to the resins at 90 °C (2 × 2 minutes). Resin cleavage process is the same as general resin cleavage[47].

For the synthesis of tri-cTat, pentaerythritol (0.5 g, 3.67 mmol, 1 equiv.) was dissolved in anhydrous DMF (20 mL). Then sodium hydride (1.76 g, 44.04 mmol, 12 equiv., 60% in oil) was added under anhydrous and anaerobic conditions at 0 °C. The mixture was allowed to react for 30 minutes. Propargyl bromide (2.4 g, 20.1 mmol, 5.5 equiv.) was subsequently added to the mixture and reacted at 0 °C for 30 min before being heated to 50 °C and reacted for 16 hours. The reaction mixture was separated between EtOAc and $H_2O$. The organic layer washed with $H_2O$ (x3) and brine (x3), then dried over anhydrous $Na_2SO_4$. The solvent was then removed in vacuo, and the product was purified by preparative TLC using 10 % EtOAc/hexane mixture. The tetrakis(2-propynyloxymethyl) methane was obtained as a white solid (890.9 mg, 3.09 mmol, 84.2 % yield). The synthesis of the azidolysine-cyclicTat peptide was completed on the Rink Amide-AM resins by introducing Fmoc-Lys(N₃)-OH and Fmoc-Lys(Mtt)-OH. The azidolysine-cyclicTat was then conjugated to the tetrakis(2-propynyloxymethyl) methane scaffold via copper-catalyzed azide alkyne cycloaddition (CuAAC) reactions. The following were dissolved in 30% DMSO/PBS: Tetrakis(2-propynyloxymethyl)methane (0.45 mg, 1.56 µmol, 1 equiv.), and azidolysine-cyclicTat (10 mg, 6.23 µmol, 4 equiv.), and $CuSO_4$ (1.56 mg, 6.23 µmol, 4 equiv.), and sodium L-ascorbate (2.5 mg, 12.64 µmol, 8 equiv.), and Tris((1-hydroxy-propyl-1H-1,2,3-triazol-4-yl)methyl)amine (THPTA) (13.6 mg, 31.2 µmol, 20 equiv.), and aminoguanidine hydrochloride (0.52 mg, 4.68 µmol, 3 equiv.) and heated with stirring at 50 °C for 30 minutes. The product was purified by HPLC (1.3 mg, 0.255 µM, 16.3% yield).

For the synthesis of biotinylated Ub-Bpa, the C-terminal glycine was coupled to Wang resin using Fmoc-Gly-OH (10 equiv.), oxyma (10.0 equiv.), DIC (10.0 equiv.), and DMAP (4-Dimethylaminopyridine, 1 equiv.) in DMF overnight at 37 °C with shaking. Bpa coupling was achieved by incubating Fmoc-BPA-OH (4 equiv.), PyAOP (4 equiv.), HOAT (4 equiv.) and NMM (N-Methylmorpholine, 8 equiv.) in DMF overnight at 37 °C with shaking. Biotin coupling was completed by incubating Biotin(5-((3aS,4S,6aR)-2-Oxohexahydro-1H-thieno[3,4-d]imidazol-4-yl) pentanoic acid, 10 equiv.), oxyma (10 equiv.), DIC (10 equiv.) in DMSO for 30 minutes at 75 °C with shaking. The remaining amino acids and AEEA were condensed using the Liberty Blue 2.0™ automated microwave peptide synthesizer.

## Preparation of $cR_{10}$-$UbcH7_{87C}$-T

$UbcH7_{87C}$ (0.5 mg/mL) was first reacted with NHS-TAMRA (8 equiv.) in PBS (pH 7.4) buffer to generate $UbcH7_{87C}$-TAMRA ($UbcH7_{87C}$-T). This fluorescent conjugate was subsequently reacted with $cR_{10}$ (3 equiv.) via a disulfide bond exchange to produce the final product ($cR_{10}$-$UbcH7_{87C}$-T), and then purified by FPLC on a Superdex 75 10/300 GL column.

## Preparation of anionic surface modified protein cargoes

The anionic peptides were first activated with 5 equiv. DTNB in reaction buffer (6 M Gn-HCl, 0.2 M NaH₂PO₄, pH 7.4) at room temperature stirring for 30 minutes. The product was then purified by RP-HPLC and lyophilized until use. Anionic peptide modified proteins with free cysteines are obtained using similar procedure, and we take the preparation of $UbcH7^{E3D2+}$-T as an example for the description. Purified UbcH7 was diluted with PBS (pH 7.4) to 0.5 mg/mL. Subsequently, 3 equiv. E3D2-TNB was added and incubated gently in a shaker at room temperature for 1 hour, followed by fluorescent labeling withing 8 equiv. NHS-TAMRA for a further 1 hour. The $UbcH7^{E3D2+}$-T product was purified by FPLC on a Superdex 75 10/300 GL column. Proteins without free cysteines such as IgG and HRP, were first thiolated with 2-iminothiolane[33], followed by disulfide bond exchange reactions to obtain the E4D3-modified proteins.

## Cellular uptake in mammalian cells

We added the final concentration of $TAT_3$ and protein cargoes directly to the cells without pre-incubation. The incubation volume of the medium is 100 µL for CLSM detection. In detail, the protein cargoes were diluted to the final concentration using DMEM, followed by the addition of $TAT_3$, then the samples were mixed and left at room temperature for 5 minutes. The cells were subsequently washed twice with PBS and then incubated with the samples at 37 °C for 30 minutes. After incubation, all samples were washed with PBS containing 25 µg/mL heparin for imaging. For live-cell confocal laser scanning microscopy (CLSM) experiments, cells were plated onto sterile 35 mm glass-bottom dishes and grown for 24 hours. Subsequently cells were incubated with corresponding samples, followed by staining with the dyes according to manufacturer's instructions and washing with PBS containing heparin for imaging.

CLSM images were taken with 63X/1.4 Oil DIC M27 lens on a Zeiss LSM 980 Axio Observer confocal laser scanning microscope. The cells were imaged using argon laser (405 nm) for visualizing Hoechst 33258 and BFP, argon laser (488 nm) for visualizing FITC, and argon laser (543 nm) for visualizing TAMRA and RFP, argon laser (639 nm) for visualizing LysoTracker™ Deep Red and SiR700-Actin. All treatments were performed in three separate experiments. All image analysis was performed using FiJi software. We used the nuclear mean fluorescence intensity (MFI) to assess the efficiency of delivery by identify individual cells and the cytosolic compartment based on Hoechst nuclear staining. The nuclear MFI is presented as the mean ± standard deviation of n = 3 independent biological replicates calculated by one-way ANOVA with Tukey's test correction. The quantification of the co-localization parameters (Pearson's and Manders' coefficients) was evaluated using the JACoP plugin in the FiJi software. In the example shown in Fig. 4d,

the magnitude of the Pearson's correlation coefficient is attributed to the local concentration of $UbcH7_{86C}^+$-T and LysoTracker™ Deep Red in the cells, which eventually results in intensity differences between the two fluorophores. The Manders' coefficient M1 represents the percentage of $UbcH7_{86C}^+$-T that overlaps with LysoTracker™ Deep Red, while the Manders' coefficient M2 represents the percentage of Lyso-Tracker™ Deep Red that overlaps with $UbcH7_{86C}^+$-T.

## Internalization mechanism study

HepG2 cells were seeded onto 6-well plates and grown for 24 hours. To investigate the internalization mechanisms, the cells were pre-incubated with various metabolic inhibitors, including 20 μM chlorpromazine (CPM), 50 μM amiloride (AM), 2.5 mM methyl-beta-dextrin (MβCD) for 30 minutes, respectively, followed by incubation with 1.5 μM $UbcH7_{86C}^+$-T and 3 μM $TAT_3$ for 30 minutes, then the cells were washed three times with PBS containing heparin. Cellular uptake was quantified by flow cytometry (CytoFLEX, Beckman) and analyzed by FlowJo software.

## Intracellular enzyme activity assay

For the RNase A activity assay, HepG2 cells were seeded in 96-well plates at 4000 cells per well for 24 hours, and followed by incubation with different concentrations of RNase A+ and RNase A in the presence of 3 μM $TAT_3$ for 30 minutes. Subsequently the cells were incubated in fresh culture medium for another 24 hours, after which CCK-8 solution was added to the wells for 2 hours and the absorbance was measured at 450 nm. For the HRP activity assay, HepG2 cells were seeded in 96-well plates at 5000 cells per well for 24 hours, and followed by incubation with 0.5 μM HRP+ and HRP in the presence of 3 μM $TAT_3$ for 30 minutes. Cellular HRP activities were then detected by TMB according to manufacturer's instructions. For the β-Gal activity assay, HepG2 cells were plated onto sterile 35 mm glass-bottom dishes and grown for 24 hours followed by incubation with corresponding samples and staining with X-Gal staining kit according to manufacturer's instructions.

## Cell viability analysis

For Cell Counting Kit-8 (CCK-8) assay, HepG2 cells were seeded in 96-well plates at 4000 cells per well for 24 hours and then treated with different concentrations of $TAT_3$ for 30 minutes. 0.64% phenol treatment was used as a control. Subsequently cells were incubated in fresh culture medium for another 24 hours and then CCK-8 solution was added to the wells for 2 hours and the absorbance was measured at 450 nm.

## Fluorescent experiments in plants

3 μM $UbcH7_{86C}^+$-T or Ub+-T was mixed with 3 μM $TAT_3$ and then loaded into a 1 mL needleless plastic syringe. Place the tip of the syringe against the back of the leaf and press the tip of the syringe lightly against the leaf and slowly depress the syringe plunger while applying counter-pressure from the other side with the index finger. Successful infiltration is observed when the water-soaked area on the leaf blade expands, and after incubation for 1 hour, the plant leaves were infiltrated with 90 μM QSY21-NHS (N-hydroxysuccinimide linked fluorescence quencher moiety) and incubated for a further 1 hour. QSY21-NHS can be used to label the primary amines ($R-NH_2$) of extracellular $UbcH7_{86C}^+$-T or $TAT_3$ to quench the extracellular fluorescence of TAMRA. The plant leaves were then cut and washed with water to remove protein samples adsorbed on the leaf surface. The leaf epidermis was torn off to allow clearer CLSM observation.

## Preparation of E2-Ub photocrosslinking probes

Biotinylated Ub31 Bpa was prepared by automated Fmoc-based microwave peptide synthesis on Wang resin and refolded in reaction buffer (20 mM Tris, 150 mM NaCl, pH 7.4). The UBE2D3 C85K S22R mutant was obtained by *E. coli* expression and purified with FPLC. To generate the E2-Ub photocrosslinking probes, the UBE2D3 mutant (200 μM) was incubated with biotinylated Ub31 Bpa (100 μM) and Uba1 (1 μM) in reaction buffer (20 mM Tris, 150 mM NaCl, 3 mM ATP, 5 mM $MgCl_2$, 1 mM TCEP, pH 8.5) at 37 °C for 24 hours. The E2-UbBpa conjugate was applied onto a Superdex ™ 75 Increase 10/300 GL column (PBS, pH 7.4) for purification. The purified probes were then conjugated with the E4D3 peptide to give E2UBD3-Ub Bpa probe+.

## Proteomic profiling of EGF-stimulated living HeLa cells

HeLa cells were first pre-incubated with MG132 and bafilomycin for 6 hours to prevent potential degradation of E3s and then stimulated with EGF. Experimental cells (A) were incubated with 8 μM probe+ and 3 μM $TAT_3$ for 30 minutes, while control cells (B) were incubated with 8 μM probe+ only. After three times washes with heparin to remove membrane-bound probes, the cells were resuspended in cold PBS and irradiated with 365 nm UV for 30 minutes at a distance of ~3 cm in cold PBS on ice to activate the probe+. Finally, the cells were lysed with RIPA lysis buffer (containing 1 mM PMSF). The lysate was centrifuged at 12 × g for 30 minutes at 4 °C, and the collected supernatant was used for subsequent proteomic analysis.

## Mass spectrometry data collection and analysis

The identification process for the E2-Ub probe was as follows: Experimental group A: 8 μM probe+ and 3 μM $TAT_3$ for 30 minutes, control group B: 8 μM probe+ only. Each group was repeated three times (biological replicates). The collected supernatants were first incubated with streptavidin agarose beads (Promega) at 4 °C overnight, then washed with four buffers in the following order: the buffer I (25 mM tris, 150 mM NaCl, 1 mM DTT, 0.5% v/v NP-40, pH = 7.6) once, the buffer II (PBS containing 0.5% w/w SDS) twice, the buffer III (PBS containing 1 M NaCl) twice, and finally the buffer IV (50 mM $NH_4HCO_3$, pH = 7.6) for ten times. The enriched proteins on the beads were then eluted with loading buffer at 95 °C for ten minutes, followed by SDS-PAGE separation. The proteins in the gel were then sequentially treated with 5 mM DTT and 11 mM IAA. In-gel digestion was then performed overnight using trypsin in a 50 mM $NH_4HCO_3$ buffer. The peptides were extracted using buffer V (a 1:1 $CH_3CN/H_2O$ buffer containing 0.1% TFA) and then concentrated.

A Thermo-Dionex Ultimate 3000 high-performance liquid chromatography system was used for gradient elution at a flow rate of 0.3 μL/min for 120 minutes. The system was connected directly to a Thermo Orbitrap Fusion 480 mass spectrometer. Separation was achieved using a custom-made fused-quartz capillary column (Upchurch, Oak Harbor, WA) with an inner diameter of 75 μm and a length of 150 mm. The column was packed with 5 μm C-18 resin (Varian, Lexington, MA) with a particle size of 300 Å. Mobile phase 1 was an aqueous solution containing 0.1% formic acid and mobile phase 2 was an acetonitrile solution containing 0.1% formic acid. Data processing utilized the Sequest HT algorithm within Proteome Discoverer software (PD, version 2.5). HeLa cell samples were analyzed using the Human Protein Database as the query source. The search parameters were set as follows: full trypsin specificity, allowing up to two uncut sites. Methionine oxidation (M) was treated as a variable modification and cysteine carbamoylation (C) as a fixed modification. The precursor ion mass tolerance was set to 20 ppm for MS scanning and 20 mmu for MS/MS spectra. The false discovery rate (FDR) for peptides was calculated using the Percolator function in the PD software with a threshold of 1% based on a decoy database.

## Statistics and reproducibility

All statistical analyses were performed using GraphPad Prism (v10.1.2). Numbers (n) of samples or replicates are indicated in the figures or figure legends. All fluorescence imaging experiments were repeated independently at least three times on different days with similar results. Control groups were included in all experiments in this study. All statistical tests performed in this study were two-sided. Statistical

significance between two groups was analyzed using an unpaired Student's *t*-test. For comparisons among multiple groups, one-way analysis of variance (ANOVA) with Tukey's test correction was used.

### Reporting summary

Further information on research design is available in the Nature Portfolio Reporting Summary linked to this article.

## Data availability

The mass spectrometry proteomics data generated in this study have been deposited in the ProteomeXchange [http://proteomecentral.proteomexchange.org] database under accession code PXD059494. Source data are provided with this paper. Data supporting the findings of this study are available from the corresponding author Yi-Ming Li upon request.

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

## Acknowledgements

This project was supported by the National Key R&D Program of China (No. 2022YFC3401500 for L. Liu), and NSFC (Nos. 22277020, 22227810 for Y.-M. Li, 22137005, 92253302 for L. Liu, 22377117 for J. Shi), and Anhui Provincial Natural Science Foundation (No. 2508085JX004 for Y.-M. Li), and the Fundamental Research Funds for the Central Universities (PA2024GDGP0037, JZ2024YQTD0600 for Y.-M. Li), and Beijing Life Science Academy (BLSA, No: 2023000CC0130) and the XPLORER prize (for L. Liu).

## Author contributions

Conceptualization and Scientific direction: Y.-M. Li, and L. Liu. Writing: Y.-M. Li, L. Liu, X. Hua. Cell biological and fluorescence imaging experiments: X. Hua, Y. Guo. Protein modification and biochemical experiments: Y. Guo, X. Han, J. Shi. Peptide synthesis: P. Li, J. Chen, J. Li, G. Chu. MS sample preparation: X. Han, Y. Wang. MS data collection and analysis: Y. Wang, Y. Guo.

## Competing interests

The authors declare no competing interests.
