## [Transparent Peer Review file · Nature Communications]

Reversible surface modifications of functional proteins for accelerated cytosolic delivery via cell-penetrating peptide clusters

Corresponding Author: Professor Lei Liu

Version 0:

Reviewer comments:

Reviewer #1

(Remarks to the Author)

In this study, the authors developed a platform for delivering diverse protein cargoes into mammalian and plant cells through a combination of anionic peptide engineering and CPP-mediated delivery. The authors are motivated by their interest in the use of E2 probes to identify their partner E3s (Ub E3 ligases). This is a very relevant scientific challenge that would have significant implications for understanding ubiquitin biology.

By conjugating proteins to a custom-designed anionic peptide (E4D3) via a cleavable disulfide bond and co-incubating with the cationic cell-penetrating peptide TAT3, they achieved efficient cytosolic delivery of a wide range of proteins with molecular weights from ~2 kDa to 430 kDa and isoelectric points from ~4 to 10. The system operates at low micromolar concentrations, avoids toxicity, and facilitates the intracellular release of native proteins upon disulfide cleavage, providing a practical and scalable strategy for functional protein delivery in both animal and plant systems. The authors have conducted a very good study, spanning development to applications. Therefore, the manuscript is suitable for Nat Com following these comments.

Comments:

1. Clarification is needed on the conjugation strategy used for attaching the cR10 CPP to the Ubch786C-T mutant, particularly since the available cysteine residue was also used to link TAMRA via a disulfide bond. However, since this protein contains only a single cysteine residue, it is unclear how both modifications were achieved simultaneously.
2. The central rationale of this study is that increasing the negative charge of protein cargo using the anionic E4D3 tag enhances electrostatic interactions with the cationic CPP TAT3, thereby promoting cellular uptake. However, if this mechanism holds, then proteins that are already highly negatively charged (i.e., low pI) should, in theory, also be internalized when co-incubated with TAT3 alone. Could the authors clarify this apparent inconsistency and explain why inherently anionic proteins failed to show delivery in the absence of the E4D3 tag? The authors are aware of the work of Liu on supercharged proteins <https://doi.org/10.1021/ja071641y>
- 3.
4. Raines and Cowrker have done a nice study on masking temporarily negative charges to make proteins permeable. What do the authors think of this in relation to their work (<https://doi.org/10.1021/jacs.7b06597>)?
5. The manuscript emphasizes delivery at "low micromolar concentrations"; however, the concentration used throughout (1.5 μ M) is generally considered moderate for CPP-based delivery, as supported by several existing studies. It would be more accurate to contextualize this concentration with relevant literature, including prior studies where these authors have shown protein delivery below one micromolar (see, for example, *Angewandte Chemie International Edition*, 2022, e2022075 and Ref. 11).
6. In this study, the anionic peptide is linked to the protein cargo via a disulfide bond, which limits this method to proteins that naturally contain cysteine residues. When cysteines are absent, thiolation is employed to introduce sulfhydryl groups; however, this chemical modification may alter the protein's structure and folding. Consequently, this approach has limitations and may not be widely applicable to all proteins. This should be discussed.
7. The authors are overlooking previous research regarding the delivery of synthetic proteins and probes. Brik et al. have

made significant contributions to the field of synthetic protein delivery, particularly in the context of the Ub and Ubl systems (e.g., Chemical Communications, 2022, 58, 8782-8785, ChemBioChem, 2022, 23, e202200122, Chemical Communications, 2021, 57, 9438 – 9441, Angewandte Chemie International Edition, 2021, 13, 60,7333 –7343, Angewandte Chemie International Edition, 2024, 63, e202410135.). In particular, some similar problems like those of Ub were used in this study to compare the delivery of different proteins and have also been reported in some of the above refs. Including the work of others not only recognizes additional contributions but also provides valuable context for readers concerning the broader landscape of protein delivery approaches and applications, as discussed in Accounts of Chemical Research (2023, 56, 14, 1953–1965). The latter reference is particularly relevant in the introduction section of the manuscript among refs 1-3 that directly refer to what is stated in the first sentences.

8. The authors should avoid overstating their findings, given that the field is already crowded with methods. Attempting to present this method as a transformative breakthrough is, at the very least, inappropriate. We have learned that every protein can yield surprises, even with minimal changes made to these proteins.

9. What is the percentage of cells that are transfected with the different proteins?

Other comments:

1. In the results section, the subtitle “UbcH786C+ is internalized via micropinocytosis” states that the internalization mechanism is via micropinocytosis; however, the authors subsequently demonstrate that macropinocytosis is the actual mode of entry. This appears to be a typographical error that should be corrected.

2. In Figure 2c, the x-axis lacks a clear label, making it unclear what the presented values represent.

3. On page 8, under the Discussion section, line 12, there appears to be a minor typographical error where “viaa” is written instead of “via.”

4. The title should exclude the concentration “Synergizing anionic chemical engineering with electrostatic interactions to empower cytosolic delivery of diverse proteins.”

Reviewer #2

(Remarks to the Author)

This is an interesting and timely contribution to the field of intracellular delivery of large biologics. The authors describe an approach to engineer the surfaces of proteins with anionic peptides, which create noncovalent electrostatic interactions with Tat clusters, and thereby deliver various protein cargoes into cells at low concentrations.

Both the use of trimeric CPP clusters and the use of non-covalent association of CPP and cargo have been demonstrated previously. Particularly, the Tat3 which the author use as their lead peptide for the delivery of cargoes into cells has been described and studied previously as 3Tat (Brock DJ, Kustigian L, Jiang M, Graham K, Wang TY, Erazo-Oliveras A, Najjar K, Zhang J, Rye H, Pellois JP. Efficient cell delivery mediated by lipid-specific endosomal escape of supercharged branched peptides. Traffic. 2018 Jun;19(6):421-35.) This article is not currently cited in the manuscript and should be included with similarities / inspiration of current work / designs clearly stated & discussed. In general, the manuscript should be amended to clarify which innovation and agents in this paper are novel, and which have been described and studied elsewhere to support the claim of novelty in this paper.

The contribution of this article to the field is the discreet engineering of proteins with anionic peptides in combination with the use of trimeric peptide clusters. This article represents an important advance of the field, and is broad in scope, exemplifying a variety of use cases which are of interest to the field of intracellular delivery.

I have several concerns about this article which should be addressed prior to publication in Nature Communications.

1) The authors mention their “standing interest in the use of E2 probes” and cite one paper [15] which appears to have originated from a different lab. Similar to general comment above, some clearer attribution of previous work is essential in revised version.

2) The description and characterization of peptide agents developed / used in this study is insufficient for reproduction – synthetic details should be added. MS analysis of peptides is very difficult to interpret, and chemical identity is hard to verify because the size of spectra is very small and m/z values cannot be discerned. Please add full size spectra.

3) Why are there cysteines in the Tat3? The original version of these peptides (Brock et al. – reference above) excludes the GC motif between the Tat sequence and the KGKGKG backbone used here. Is there a particular reason why these cysteines were added? These could form disulphide bonds among themselves or with cargoes – this could fundamentally change the mechanism of cargo-CPP interaction proposed here. Also – is the C-terminal of the backbone peptide a ketone or is that a typo?

4) The structure of the tri-cTat CPP is different from the agent disclosed in the literature. Is that an intentional change? Compared to original, there is a CH₂ group missing in the central scaffold. Synthetic method should be added to manuscript (particularly if different from original molecule).

5) Manuscript requires further information on quantification. N numbers of individual cells analysed should be quoted for all analysis; the microscopy data in the majority of extended figures has not been analysed / quantified.

6) Quantification of delivery – Nuclear MFI – is this the most appropriate measure of efficacy given that the proposed mechanism of delivery is that anionic surface modification and a CPP detach from cargo upon delivery into the cytosol. Do protein cargoes traffic into the nucleus without CPP?

7) All optimization of this technology was done on one specific enzyme. The paper makes a claim for this approach as generally applicable technology; while the delivery of other cargoes is presented in Figure 4, it is not possible to establish the efficacy of delivery from the data presented. Efficacy of uptake data should be added for these cargoes or claim at generality

revised and discussed appropriately in the Discussion section.

Reviewer #3

(Remarks to the Author)

Major Concerns

- o The manuscript presents the biotinylated E2~Ub photocrosslinking probe as if it were an innovation of the authors. However, the precise probe architecture has previously been reported (Mathur et al.). Proper scholarly practice requires the authors to cite this prior work explicitly at the appropriate point in the manuscript and clarify that their contribution is the synthesis of a variant carrying the E4D3 tag.
- o Details on probe synthesis are insufficient. It is unclear how biotin was introduced into the probe and the precise chemical structure of the biotin + linker region probe is not described. Moreover, no mass spectrometry data are provided to confirm the integrity and identity of the final construct.
- o The authors claim successful probe delivery into cells, but the images suggest that the probe does not distribute diffusely in all cells. Instead, it accumulates in subcellular puncta in some cells. This observation must be clarified, as it has important implications for probe E3 accessibility, target engagement, and interpretation of the results.
- o The experiment replicating Mathur et al., using lysates, with EGF treatment lacks a minus-EGF control. I am not suggesting the experiment be repeated, but the purpose of EGF treatment in Mathur et al. was to demonstrate activation of CBL. PJA2 was also detected upon EGF stimulation, but neither are in the current study highlighting a shortcoming with the approach for studying these EGF responsive E3s. This should be discussed and reconciled. A likely explanation is the subcellular location of the probe is restricted, as suggested by the microscopy.
- o The manuscript misrepresents prior work by implying that only 7 RING E3 ligases were detected by Mathur et al. when in fact 25 were reported. It states that 4 of the 7 are unique to Mathur et al. This is incorrect as the number is far higher.
- o Mathur et al. restricted their analysis to proteins annotated with the Pfam term "RING". This excluded cullin scaffolds, substrate receptors, the exchange factor CAND1, and UBR4. When this is considered, the number of detected RING E3s in the authors study is ~26. For an accurate comparison, the available raw data from Mathur might need to be subjected to be re filtering because the Pfam database would have been updated since that study.
- o Taken together, the current comparative analysis is flawed. The reality is that the in-cell and lysate-based approaches are complementary with a similar number of uniquely detected RING E3s, and partial overlap. This must be revised.

Summary

The manuscript requires significant revision for scholarly accuracy, methodological transparency, and fair contextualisation. Specifically:

- Cite prior work correctly and position the present study as a variant approach.
- Provide clear methodological details (biotin incorporation, probe structure, mass spec validation).
- Address conclusions drawn about probe subcellular localisation.
- Correct the comparative analysis to reflect that the two approaches are complementary rather than the careful reanalysis of the unique and overlapping E3s after filtering for RING E3s.

Version 1:

Reviewer comments:

Reviewer #1

(Remarks to the Author)

The authors have done very good work on revising the current MS. Just one note that the cited account of chemical research appears in Ref 3, should be the following one and not the one that appears in this version:

Cite this: Acc. Chem. Res. 2022, 55, 15, 2055–2067
<https://doi.org/10.1021/acs.accounts.2c00236>

Assessment of authors' responses to comments of Reviewer 2:

Overall, the revision is excellent. However, the following correction should be done before recommending publication.

Comment # 1: The author should cite *Angewandte Chemie*, 2018, International Edition, 57, 5645-5649, as the original work on the sequential DHA formation, not the cited paper in ref 15.

Reviewer #3

(Remarks to the Author)

The authors have addressed my concerns.

Reviewer 1:

In this study, the authors developed a platform for delivering diverse protein cargoes into mammalian and plant cells through a combination of anionic peptide engineering and CPP-mediated delivery. The authors are motivated by their interest in the use of E2 probes to identify their partner E3s (Ub E3 ligases). This is a very relevant scientific challenge that would have significant implications for understanding ubiquitin biology.

By conjugating proteins to a custom-designed anionic peptide (E4D3) via a cleavable disulfide bond and co-incubating with the cationic cell-penetrating peptide TAT3, they achieved efficient cytosolic delivery of a wide range of proteins with molecular weights from ~2 kDa to 430 kDa and isoelectric points from ~4 to 10. The system operates at low micromolar concentrations, avoids toxicity, and facilitates the intracellular release of native proteins upon disulfide cleavage, providing a practical and scalable strategy for functional protein delivery in both animal and plant systems. The authors have conducted a very good study, spanning development to applications. Therefore, the manuscript is suitable for Nat Com following these comments.

Response: We thank for the positive comments.

Comments:

1. Clarification is needed on the conjugation strategy used for attaching the cR10 CPP to the UbcH786C-T mutant, particularly since the available cysteine residue was also used to link TAMRA via a disulfide bond. However, since this protein contains only a single cysteine residue, it is unclear how both modifications were achieved simultaneously.

Response: As shown in **Figure 1** below, UbcH_{787C} was first labeled with NHS-TAMRA via a nucleophilic substitution reaction to generate UbcH_{787C}-TAMRA (UbcH_{787C}-T). This fluorescent conjugate was subsequently reacted with cR10 via a disulfide bond exchange to produce the final product, cR₁₀-UbcH_{787C}-T. We have provided a detailed description of the sample preparation process in the revised manuscript (P20, line 471-475) and in Extended Data Fig. 1B.

Figure 1. Preparation flowchart of cR₁₀-UbcH_{787C}-T.

2. The central rationale of this study is that increasing the negative charge of protein cargo using the anionic E4D3 tag enhances electrostatic interactions with the cationic CPP TAT3, thereby promoting cellular uptake. However, if this mechanism holds, then proteins that are already highly negatively charged (i.e., low pI) should, in theory, also be internalized when co-incubated with TAT3 alone. Could the authors clarify this apparent inconsistency and explain why inherently anionic proteins failed to show delivery in the absence of the E4D3 tag? The authors are aware of the work of Liu on supercharged proteins <https://doi.org/10.1021/ja071641y>.

Response: The central idea of this work is to carry out “**localized** charge engineering” on protein surfaces, thereby artificially inducing electrostatic interactions with cationic cell-penetrating peptides (CPPs) to achieve efficient cytosolic protein delivery. Although the isoelectric points (pIs) of proteins such as bovine serum albumin (BSA) and blue fluorescent protein (BFP) are relatively low — approximately 4.7 for BSA and 6.6 for BFP — the pIs value reflects the average charge density of the entire protein structure. This average negative charge distribution may not be sufficient to form high-density **local negative charge "patches"** capable of generating electrostatic interactions with TAT₃. Our experimental data (**Figure 2**) also support this explanation,

that is, neither native BSA nor BFP alone can be successfully delivered into cells via co-incubation with TAT₃. In contrast, the E4D3 modification introduces a high-density anionic peptide segment at specific sites on the protein surface, thereby creating a localized negatively charged microenvironment. This design allows for strong electrostatic association with the positively charged TAT₃, which is crucial for the successful delivery in this study (Page 10, lines 287-292).

We thank the reviewer for drawing our attention to the work of David R. Liu's team on supercharged proteins (*J. Am. Chem. Soc.* **2007**, 129, 10110). Compared with the strategy of regulating the overall net charge of proteins in that work, our method focuses more on engineering local charge density. This provides a valuable approach for delivering proteins that are inherently difficult to convert into a "supercharged" state. We have now cited this important Ref. in the "Introduction" section (Ref. 6) of the revised manuscript.

Figure 2. Cellular uptake of BSA and BFP using TAT₃.

3. Raines and Coworker have done a nice study on masking temporarily negative charges to make proteins permeable. What do the authors think of this in relation to their work (<https://doi.org/10.1021/jacs.7b06597>)?

Response: Although both our and Raines's strategy focus on "charge engineering" on the protein surfaces, the Raines's approach employs multi-site chemical modification to mask the native negative charges on protein cargos, such as GFP (**Figure 3**). This increases their net positive charge, enabling autonomous cellular uptake. In contrast, our strategy involves site-specific conjugation of the E4D3 peptide to increase **local** negative charge density. This facilitates **localized electrostatic interactions** with the cationic CPP TAT₃, thereby achieving TAT₃-mediated transmembrane delivery.

We thank the reviewer for this valuable Ref. and have cited it in the relevant section of the manuscript (see Ref. 51).

Figure 3. Comparison of these two approaches

5. *The manuscript emphasizes delivery at "low micromolar concentrations"; however, the concentration used throughout (1.5 μM) is generally considered moderate for CPP-based delivery, as supported by several existing studies. It would be more accurate to contextualize this concentration with relevant literature, including prior studies where these authors have shown protein delivery below one micromolar (see, for example, Angewandte Chemie International Edition, 2022, e2022075 and Ref. 11).*

Response: We agree with the reviewer's comment and have replaced the term "low concentration" with the specific value of "1.5 μM " throughout the manuscript.

6. *In this study, the anionic peptide is linked to the protein cargo via a disulfide bond, which limits this method to proteins that naturally contain cysteine residues. When cysteines are absent, thiolation is employed to introduce sulfhydryl groups; however, this chemical modification may alter the protein's structure and folding. Consequently, this approach has limitations and may not be widely applicable to all proteins. This should be discussed.*

Response: We thank this suggestion. Although previous studies (*Nat. Chem.* **2021**, *13*, 530) have indicated that introducing sulfhydryl groups through thiolation may not affect the function of certain proteins, we agree with the reviewer that such chemical modifications could potentially alter the structure and bioactivity of other proteins. Accordingly, we have included additional analysis addressing this point in the "Discussion" section (Page 10, lines 276-278) of the revised manuscript.

7. *The authors are overlooking previous research regarding the delivery of synthetic proteins and probes. Brik et al. have made significant contributions to the field of synthetic protein delivery, particularly in the context of the Ub and Ubl systems (e.g., Chemical Communications, 2022, 58, 8782-8785, ChemBioChem, 2022, 23, e202200122, Chemical Communications, 2021, 57, 9438 - 9441, Angewandte Chemie International Edition, 2021, 13, 60,7333 -7343, Angewandte Chemie International Edition, 2024, 63, e202410135.)*

In particular, some similar problems like those of Ub were used in this study to compare the delivery of different proteins and have also been reported in some of the above refs. Including the work of others not only recognizes additional contributions but also provides valuable context for readers concerning the broader landscape of protein delivery approaches and applications, as discussed in Accounts of Chemical Research (2023, 56, 14, 1953–1965). The latter Ref. is particularly relevant in the introduction section of the manuscript among refs 1-3 that directly refer to what is stated in the first sentences.

Response: We apologize for the missing of these significant contributions from Brik's group. We have now incorporated key publications from the Brik group on ubiquitin delivery (*Angew. Chem. Int. Ed.* **2021**, *60*, 7333-7343. See Ref. 7; *Angew. Chem. Int. Ed.* **2024**, *63*, e202410135. See Ref. 5), as well as a highly relevant review article (*Acc. Chem. Res.* **2023**, *56*, 1953-1965. See Ref. 3) in revised manuscript.

8. *The authors should avoid overstating their findings, given that the field is already crowded with methods. Attempting to present this method as a transformative breakthrough is, at the very least, inappropriate. We have learned that every protein can yield surprises, even with minimal changes made to these proteins.*

Response: We have checked the entire manuscript thoroughly and removed the overstated claims.

9. *What is the percentage of cells that are transfected with the different proteins?*

Response: The transduction efficiency of proteins was quantified using a combination of flow cytometry and confocal fluorescence imaging. Flow cytometric analysis of UbcH7_{86C}-T showed a transduction efficiency of 72.4% in HepG2 cells when incubated with of 1.5 μM UbcH7_{86C}-T and 3 μM TAT₃ (**Figure 4A**). To validate this finding, we further analyzed fluorescence images from three independent experiments using the statistical method described in the literature (*Nat. Chem.* **2022**, *14*, 284-293) (**Figure 4B**), which yielded a closely matched transduction efficiency of 77.8% (**Figure 4C**).

We then used the image-based analytical approach to evaluate the delivery efficiency of other fluorescently labeled proteins. As summarized in **Figure 4C**, all of the proteins exhibited consistent transduction efficiencies exceeding 55%. These quantitative efficiency data are now presented as a new bar graph in Fig. 4D in the manuscript, and representative fluorescence images for these experiments can be found in the Supporting Information (Fig. S4).

For non-fluorescent functional proteins (e.g., RNase A and HRP) whose transduction efficiency cannot be directly measured via flow cytometry or imaging, their successful delivery has also been demonstrated through intracellular functional assays (see Fig. 4E-G in the revised manuscript). We have strengthened the analysis and discussion of transduction efficiencies across different protein types in the revised manuscript.

Figure 4. (A) Flow cytometry analysis of the HepG2 cells treated with 1.5 μM UbcH7_{86C}⁺-T in the presence of TAT₃ for 30 minutes. (B) Quantification methods of the percentage of transduced cells in the literature (*Nat. Chem.* **2022**, *14*, 284-293). (C) Quantification of the percentage of transduced cells in the manuscript.

Other comments:

1. In the results section, the subtitle "UbcH786C+ is internalized via micropinocytosis" states that the internalization mechanism is via micropinocytosis; however, the authors subsequently demonstrate that macropinocytosis is the actual mode of entry. This appears to be a typographical error that should be corrected.

Response: We have corrected "micropinocytosis" to "macropinocytosis" throughout the manuscript.

2. In Figure 2c, the x-axis lacks a clear label, making it unclear what the presented values represent.

Response: We have added a label above the graph in Figure 2c in the revised manuscript.

3. On page 8, under the Discussion section, line 12, there appears to be a minor typographical error where "viaa" is written instead of "via."

Response: We have corrected "viaa" to "via" in the manuscript.

4. The title should exclude the concentration " Synergizing anionic chemical engineering with electrostatic interactions to empower cytosolic delivery of diverse proteins."

Response: In accordance with the reviewer's suggestion, we have revised the title of the manuscript, especially removing the descriptor "low micromolar concentrations."

Reviewer 2:

This is an interesting and timely contribution to the field of intracellular delivery of large biologics. The authors describe an approach to engineer the surfaces of proteins with anionic peptides, which create noncovalent electrostatic interactions with Tat clusters, and thereby deliver various protein cargoes into cells at low concentrations.

*Both the use of trimeric CPP clusters and the use of non-covalent association of CPP and cargo have been demonstrated previously. Particularly, the Tat3 which the author use as their lead peptide for the delivery of cargos into cells has been described and studied previously as 3Tat (Brock DJ, Kustigian L, Jiang M, Graham K, Wang TY, Erazo - Oliveras A, Najjar K, Zhang J, Rye H, Pellois JP. Efficient cell delivery mediated by lipid - specific endosomal escape of supercharged branched peptides. *Traffic*. 2018 Jun;19(6):421-35.)*

This article is not currently cited in the manuscript and should be include with similarities / inspiration of current work / designs clearly stated & discussed. In general, the manuscript should be amended to clarify which innovation and agents in this paper are novel, and which have been described and studied elsewhere to support the claim of novelty in this paper.

The contribution of this article to the field is the discreet engineering of proteins with anionic peptides in combination with the use of trimeric peptide clusters. This article represents an important advance of the field, and is broad in scope, exemplifying a variety of use cases which are of interest to the field of intracellular delivery.

Response: We thank the reviewer for their positive comments. We have carefully revised the manuscript according to suggestions, with specific revision and clarifications provided below:

(1) In the section of TAT₃ synthesis, we have now cited the foundational work by Brock et al. (*Traffic*. 2018 Jun;19(6):421-3; see Ref. 18) and clarified that the TAT₃ used in this study is based on the supercharged branched peptides originally reported by Brock et al (Page 4, lines 73-75).

(2) As rightly noted by the reviewer, the core innovation of our work not lies in TAT₃ itself, but in the introduction of a novel strategy that enables the efficient and general cytosolic protein delivery by engineering protein cargos with anionic motifs and using trimeric peptide clusters as carriers. In this context, E4D3 is a newly designed peptide. Conjugation of this peptide to protein cargos in a site-specific manner creates a localized, high-density negative surface charge, which significantly enhances electrostatic interaction with TAT₃. This ultimately enables the efficient delivery of a broad range of proteins that were previously difficult to transport. We have now articulated this innovative aspect in both the "Results" and "Discussion" sections (Page 10, lines 271-274) more clearly to better distinguish our contribution from prior studies.

I have several concerns about this article which should be addressed prior to publication in Nature Communications.

Comments:

1. The authors mention their "standing interest in the use of E2 probes" and cite one paper [15] which appears to have originated from a different lab. Similar to general comment above, some clearer attribution of previous is essential in revised version.

Response: We thank the reviewer for this observation. A long-standing interest of our laboratory is to develop E2-based probes to elucidate the functions of their E3 partners, as demonstrated by our previous work (*Chem. Commun.* 2019, 55, 7109; *Angew. Chem. Int. Ed.* 2021, 60, 17171; *Sci. China Chem.*, 2025, 68. <https://doi.org/10.1007/s11426-025-2697-6>). In revised manuscript, we have updated the Ref. 15 where we describe this "standing interest" (*Angew. Chem. Int. Ed.* 2021, 60, 17171).

2. The description and characterization of peptide agents developed / used in this study is insufficient for reproduction – synthetic details should be added. MS analysis of peptides is very difficult to interpret, and chemical identity is hard to verify because the size of spectra is very small and m/z values cannot be discerned.

Please add full size spectra.

Response: We have added the synthetic methods and purification procedures for all peptides in "Methods" section. We also have provided the high-resolution mass spectrometry data in Fig. S1, where all m/z values are clearly labeled.

3. Why are there cysteines in the Tat3? The original version of these peptides (Brock et al. – Ref. above) excludes the GC motif between the Tat sequence and the KGKGKG backbone used here. Is there a particular reason why these cysteines were added? These could form disulphide bonds among themselves or with cargos – this could fundamentally change the mechanism of cargo-CPP interaction proposed here. Also – is the C-terminal of the backbone peptide a ketone or is that a typo?

Response: In our initial approach to synthesize TAT₃, we used native chemical ligation (NCL) for a two-fragment ligation (Figure 5A). Specifically, cysteine was conjugated to the lysine side chains of a KGKGKG backbone, which was then ligated with GRKKRRQRRRG-hydrazide via NCL. However, this method yielded an efficiency of only 12.2%. Following systematic optimization, we found that performing linear solid-phase peptide synthesis of TAT₃ using Rink Amide Resin could achieve a notably higher yield of 55.2%. As a result, the C-terminus of the resulting TAT₃ is an amide group (–CONH₂).

Chromatographic and mass spectrometric analyses revealed no detectable disulfide-bonded products, either for TAT₃ alone (Figure 5B) or when mixed with the cargo protein UbcH7_{86C}⁺ (Figure 5C). Furthermore, native gel electrophoresis confirmed the formation of electrostatic interactions between TAT₃ and UbcH7_{86C}⁺ after mixing (Figure 5D). These observations are consistent with the conclusion reported in the literature that "The presence of the thiols is not necessary to achieve delivery" (*Bioconjugate Chem.* 2010, 21, 2164; P3, left panel, line 36-37).

Figure 5. (A) Synthesis route of TAT₃. (B) Chromatographic and mass spectrometric characterisation of TAT₃. (C) Stability evaluation of mixture of UbcH7_{86C}^{E4D3+} and TAT₃. The mixture was preserved at room temperature and detected by LC-MS for 16 days. (D) Coomassie blue staining and fluorescence images of UbcH7_{86C}⁺-T and TAT₃ samples in the presence of different concentrations of heparin.

4. The structure of the tri-cTat CPP is different from the agent disclosed in the literature. Is that an intentional

change? Compared to original, there is a CH₂ group missing in the central scaffold. Synthetic method should be added to manuscript (particularly if different from original molecule).

Response: We thank the reviewer for pointing out this mistake. The tri-cTat molecule used in our study is exactly the same as that reported in the literature (*Nat. Chem.* **2022**, *14*, 284). We have corrected the molecular structure in Fig. S1B of the revised manuscript and added a detailed description of the synthetic procedure in the "Methods" section (Page 19, lines 429-446).

5. Manuscript requires further information on quantification. N numbers of individual cells analysed should be quoted for all analysis; the microscopy data in the majority of extended figures has not by analysed / quantified.

Response: We have revised all figure captions involving quantitative analysis of microscopy images, explicitly quoted the individual cell number for all analysis. Corresponding statistical details have also been updated in the "Methods" section.

The key image data in the Extended Data Figures has now been analyzed quantitatively and integrated into the main text. Specifically, the quantification of Extended Data Fig. 3A corresponds to Fig. 2C, the quantification of Extended Data Fig. 3C corresponds to Fig. 2D, and quantification of Extended Data Fig. 8A is now presented in Extended Data Fig. 8B. Quantitative analysis of some earlier images (e.g., Extended Data Fig. 1D, 1E, 1F, and 2B) from preliminary CPP screening experiments, quantitative analysis was deemed unreliable due to generally weak fluorescence signals and low signal-to-noise ratios. These images are retained as qualitative examples, and their captions have been updated to clarify their illustrative purpose and the reason why they have not been quantified (due to unsuitable signal characteristics).

6. Quantification of delivery – Nuclear MFI – is this the most appropriate measure of efficacy given that the proposed mechanism of delivery is that anionic surface modification and a CPP detach from cargo upon delivery into the cytosol. Do protein cargos traffic into the nucleus without CPP?

Response: (1) We selected nuclear MFI as a quantitative metric because it can distinguish between proteins that have successfully escaped into the cytosol and those that are trapped within endosomes. Only soluble proteins that have completed endosomal escape and exhibit diffuse distribution throughout the cell (such as UbcH7_{86C} used in this manuscript) can freely diffuse into the nucleus (see, for example, the ubiquitin delivery in **Figure 6A** of *Angew. Chem. Int. Ed.* **2021**, *60*, 7333). This approach thus provides a more accurate measure of cytosolic delivery efficiency (**Figure 6B**, *Nat. Chem.* **2021**, *13*, 530). This quantification method has been adopted in previous studies (*Nat. Chem.* **2021**, *13*, 530; *J. Am. Chem. Soc.* **2023**, *145*, 24535), and we have now clarified its rationale in the revised manuscript (Page 4, lines 92-93).

(2) It has been reported that UbcH7 itself can freely diffuse through nuclear pores independently of CPPs and exhibits a diffuse distribution throughout the entire cell (**Figure 6C**, *J. Biol. Chem.* **2001**, *276*, 19640; *J. Endocrinol.* **2006**, *190*, 621). Our experimental results are consistent with this: red fluorescently labeled UbcH7_{86C} exhibited a diffuse distribution throughout the entire cell (**Figure 6D**), whereas green fluorescently labelled TAT₃ accumulated in the nucleoli (**Figure 6E**). This spatial segregation suggests that UbcH7_{86C} dissociates from TAT₃ in the cytosol. Therefore, the nuclear fluorescence signal of UbcH7_{86C} is derived from the dissociated UbcH7_{86C} that has diffused freely into the nucleus.

Figure 6 (A) Nuclear MFI of ubiquitin in the literature (*Angew. Chem. Int. Ed.* **2021**, *60*, 7333-7343). (B) Nuclear MFI of NLS-mCherry in the literature (*Nat. Chem.* **2021**, *13*, 530-539). (C) Fluorescence images of COS-7 cells transfected with UbcH7-GFP in the literature (*J. Biol. Chem.* **2001**, *276*, 19640). (D) Quantification of the UbcH7_{86C}-T fluorescence intensity along the white lines shown in the left-hand fluorescence images. The gray box denotes the nuclear region. (E) Quantification of the FITC-TAT₃ fluorescence intensity along the white lines shown in the left-hand fluorescence images. The gray box denotes the nuclear region.

7. All optimization of this technology was done on one specific enzyme. The paper makes a claim for this approach as generally applicable technology; while the delivery of other cargos is presented in Figure 4, it is not possible to establish the efficacy of delivery from the data presented. Efficacy of uptake data should be added for these cargos or claim at generality revised and discussed appropriately in the Discussion section.

Response: Following the established statistical approach from *Nat. Chem.* **2022**, *14*, 284 (**Figure 7A**), we quantified the transduction efficiency of all fluorescently labeled protein cargos by analyzing fluorescence images from three independent biological replicates. As shown in **Figure 7B**, the calculated transduction efficiency for UbcH7_{86C}-T was 77.8%, which closely matches the 72.4% positive cell rate determined by flow cytometry, confirming the reliability of our measurements.

Using this image-based quantification method, we found that the delivery efficiencies of other fluorescently labeled proteins (Lifeact, NLS-UbcH7, BFP, IgG, RFP, BSA) all exceeded 55% (**Figure 7B**). These quantitative efficiency data are now presented as a new bar graph in Fig. 4D in the manuscript (Page 7, lines 192-193), and representative fluorescence images for these experiments can be found in the Supporting Information (Fig. S4). For non-fluorescent functional proteins (e.g., RNase A and HRP) whose transduction efficiency cannot be directly measured via flow cytometry or imaging, their successful delivery has also been demonstrated through intracellular functional assays (see Fig. 4E-G in the revised manuscript).

In addition, we have carefully moderated the claims regarding "generality" throughout the revised manuscript, and focused on the demonstrated capacity to deliver multiple distinct protein cargos in the discussion section.

Figure 7 (A) Quantification methods of the percentage of transduced cells in the literature (*Nat. Chem.* **2022**, *14*, 284-293). (B) Quantification of the percentage of transduced cells in the manuscript.

Reviewer #3 (Remarks to the Author):

Major Concerns

1. The manuscript presents the biotinylated E2~Ub photocrosslinking probe as if it were an innovation of the authors. However, the precise probe architecture has previously been reported (Mathur et al.). Proper scholarly practice requires the authors to cite this prior work explicitly at the appropriate point in the manuscript and clarify that their contribution is the synthesis of a variant carrying the E4D3 tag.

Response: We agree with the reviewer's opinion. The structure of the biotinylated E2-Ub probe used in our study originates from the work of Mathur et al. (*Cell Chem. Biol.* **2020**, 27, 74-82. e6.). Our contribution is chemically conjugating an E4D3 tag to this probe, which enables its delivery into living cells. We have now cited this foundational work in the section of probe synthesis and reclassified our study as a complementary approach based on their design (Page 9, lines 239-240).

2. Details on probe synthesis are insufficient. It is unclear how biotin was introduced into the probe and the precise chemical structure of the biotin + linker region probe is not described. Moreover, no mass spectrometry data are provided to confirm the integrity and identity of the final construct.

Response: The biotinylated Ub-Bpa was prepared via linear solid-phase peptide synthesis, with two AEEA groups incorporated between the biotin and ubiquitin sequences (**Figure 8A**, according to *JACS Au* **2023**, 3, 2873). We have provided mass spectrometry of the final product E2-Ub⁺ probe (**Figure 8B**, also see Fig. S2 in the revised manuscript), and the detailed synthetic procedure has been updated in the "Methods" section.

Figure 8 (A) Sequence and chemical structure of biotinylated Ub-Bpa. (B) Mass spectrometric of E2-Ub⁺ probe. Due to the presence of three free cysteine residues (C21, C107 and C111) on UBE2D3-Ub, the molecular weight of 30,762 corresponds to the final product E2-Ub⁺ probe (UBE2D3-Ub conjugated with three E4D3 tags). Molecular weights of 28,240 Da and 29,501 Da correspond to minor products featuring conjugation with one or two E4D3 tags, respectively.

3. The authors claim successful probe delivery into cells, but the images suggest that the probe does not distribute diffusely in all cells. Instead, it accumulates in subcellular puncta in some cells. This observation must be clarified, as it has important implications for probe E3 accessibility, target engagement, and interpretation of the results.

Response: We agree with the reviewer that punctate fluorescence patterns were indeed observed in some cells, indicating probe accumulation within the endosome. Colocalization analysis using LysoTracker revealed that more than 70% of the internalized probe successfully reached the cytosol via endosomal escape pathways (**Figure 9A**). This finding is consistent with the previous reports demonstrating that a fraction of CPP-delivered protein cargos, particularly larger macromolecules such as mCherry and Fab fragments (*Nat. Chem.* **2021**, 13,

530; *Nat. Chem.* 2022, 14, 284), typically remain sequestered within lysosomes and appear as bright puncta (**Figure 9B**).

Meanwhile, proteomic analysis further confirms that the probe remains functionally active when delivered into live cells, as demonstrated by its ability to capture known UBE2D3 partner E3s (Fig. 6D in the revised manuscript).

Figure 9 (A) Whole cell co-localization analysis of Probe⁺-T and LysoTracker signal. Co-localization images from the left: red channel (Probe⁺-T); purple channel (LysoTracker). (B) Representative images of cytosolic delivery of mCherry (*Nat. Chem.* **2021**, 13, 530-539) and Fab (*Nat. Chem.* 2022, 14, 284-293.).

4. The experiment replicating Mathur et al., using lysates, with EGF treatment lacks a minus-EGF control. I am not suggesting the experiment be repeated, but the purpose of EGF treatment in Mathur et al. was to demonstrate activation of CBL. PJA2 was also detected upon EGF stimulation, but neither are in the current study highlighting a shortcoming with the approach for studying these EGF responsive E3s. This should be discussed and reconciled. A likely explanation is the subcellular location of the probe is restricted, as suggested by the microscopy.

Response: Upon careful re-examining our proteomics dataset, we noted that the two E3s (CBL and PJA2) mentioned by the reviewer were also identified in our experiments (**Table 1**). However, based on our predefined thresholds, they were not classified as significantly enriched (proteins with a fold change > 2 and a p-value < 0.05 were considered as significantly enriched). As Fig. 6D in the main text only displays significantly enriched proteins, CBL and PJA2 were excluded.

We also agree with the reviewer's opinion that the partial entrapment of the delivered probes within endosomal compartments may have restricted their access to certain E3s spatially, thereby limiting their capture. As suggested, we have now added a discussion of this point in the corresponding section (Page 9, lines 248-250).

C: Accession	Gene name	Abundances: C1 Sample	Abundances: C2 Sample	Abundances: C3 Sample	Abundances: S1 Sample	Abundances: S2 Sample	Abundances: S3: Sample
P22681	CBL	266996	634300	292309	337656	1241400	261477
O43164	PJA2	377155	121890	279441	180775	175399	214645

Table 1 Raw data of CBL and PJA2 identified in this manuscript.

5. The manuscript misrepresents prior work by implying that only 7 RING E3 ligases were detected by Mathur et al. when in fact 25 were reported. It states that 4 of the 7 are unique to Mathur et al. This is incorrect as the number is far higher.

Response: In the original manuscript, our statement was: "Notably, the UBE2D3-Ub Bpa probe was previously used to profile RING E3s in EGF-stimulated HEK293T cell lysates. A comparison of these two data sets (Table S3) revealed that 7 partner E3s were labeled in both living cells and lysates,". This intended to indicate that seven

UBE2D3 partner E3s identified by our method overlapped with those identified by Mathur et al., rather than that only 7 RING E3s were detected in their work. To prevent any potential misinterpretation, we have deleted this sentence from the revised manuscript.

6. Mathur et al. restricted their analysis to proteins annotated with the Pfam term “RING”. This excluded cullin scaffolds, substrate receptors, the exchange factor CAND1, and UBR4. When this is considered, the number of detected RING E3s in the authors study is ~26. The an accurate comparison, the available raw data from Mathur might need to be subjected to be re filtering because the pram domain database would have been updated since that study.

Taken together, the current comparative analysis is flawed. The reality is that the in-cell and lysate-based approaches are complementary with a similar number of uniquely detected RING E3s, and partial overlap. This must be revised.

Response: As requested by the reviewer, we carefully analyzed our experimental data. After excluding the cullin scaffolds, substrate receptors, the exchange factor CAND1, and UBR4, we identified 42 RING-type E3 ligases totally, among which 21 were significantly enriched (fold change > 2 and p-value < 0.05).

Comparing this dataset with that of Mathur et al., we found that 10 of these E3 ligases were identified by both methods: BRAP, CBL, PJA2, MIB1, RAD18, RING1, RNF168, RNF10, RNF114, and RNF126. We also identified additional E3s, such as UHRF1, TRIM32, ZNF598, RNF168 and RNF181, and Mathur et al. also reported different E3s, including BRCA1, BRE1A, BRE1B, HLTF and PCGF6. Together, these findings collectively show the complementarity between the two approaches, offering valuable tools for studying E2–E3 pairing relationships.

We have updated Figure 6D, Table S1 and Table S2 in the manuscript accordingly, and revised the corresponding descriptions and analyses in the main text (Page 9, lines 263-267).

Summary

The manuscript requires significant revision for scholarly accuracy, methodological transparency, and fair contextualisation. Specifically:

- Cite prior work correctly and position the present study as a variant approach.*
- Provide clear methodological details (biotin incorporation, probe structure, mass spec validation).*
- Address conclusions drawn about probe subcellular localisation.*
- Correct the comparative analysis to reflect that the two approaches are complementary rather with the careful reanalysis of the unique and overlapping E3s after filtering for RING E3s.*

Response: We have made revision to the four aspects of suggestions by the reviewers in combination with each of the above specific issues.

We greatly thank these referees for the insightful suggestions that greatly help us improve the quality of the paper.

With my best wishes,

Lei Liu, Ph.D.